# Learning Structured Set Utility Functions with Contrastive Element Representations

## Abstract

Learning utility functions over sets of elements is central to many machine learning and decision-making tasks such as feature selection, sensor placement, and content recommendation, where the goal is to evaluate and select an optimal subset of elements that provide the largest utility. These utility functions often exhibit desirable properties like monotonicity and submodularity over sets, but are typically expensive to evaluate and may lack an explicit analytical form. Moreover, the utility of a set can vary depending on certain contextual variables, further complicating the learning task. In this work, we propose a unified framework for modeling and learning contextual set functions with monotone submodular structure from data using deep networks equipped with structural regularization. Our key insight is to decompose the set function into two learnable components: (i) a context-conditioned contrastive embedding network that maps elements to a shared latent space based on performance and contextual similarity, and (ii) an aggregation network that predicts set-level utility from the sum of embeddings with a submodular norm-based regularization term encouraging the learned function to exhibit diminishing returns. This combination improves utility prediction for unseen sets and contexts and enables greedy subset selection, which admits near-optimality guarantees. We evaluate our framework on a wide variety of real-world contextual subset selection tasks such as content recommendation, document summarization, and sensor selection demonstrating consistent improvements in utility prediction compared to baselines and stronger subset selection performance under context shifts.

## 1 Introduction

Many tasks in machine learning and decision-making require selecting subsets of items that jointly maximize a desired utility. These tasks often arise in domains such as sensor selection Krause et al. (2008); Guestrin et al. (2005); Gottipati et al. (2019), feature acquisition Kontschieder et al. (2015); Janisch et al. (2020); Shim et al. (2018), experiment design Chaloner & Verdinelli (1995); Krause et al. (2008); Hebbal et al. (2023), resource allocation Tsitsiklis (1993); Buchgraber & Shutin (2012); Efroni et al. (2018), and recommender systems Mehrotra & Vishnoi (2023); Wang et al. (2020); Qin & Zhu (2013), where one must choose a subset of elements (e.g., sensors, features, samples) under budget or operational constraints. A common trait in these problems is that the utility of a set exhibits diminishing returns, i.e., adding an element to a smaller set provides more gain than adding it to a larger one (which is a superset of the smaller set). Such structure is naturally modeled by submodular set functions Nemhauser et al. (1978), which admit efficient approximation algorithms for subset selection via greedy strategies with near-optimality guarantees.

However, in many real-world applications, the true utility function may be unknown, may not admit a closed form analytical expression, or may be expensive to evaluate. For instance, in environmental monitoring or surveillance, the utility of a sensor configuration may depend on complex interactions between sensor specifications and environmental conditions. Moreover, the underlying utility may be contextual, varying across different deployment environments or tasks. In such cases, evaluating the true utility for all possible subsets and contexts is infeasible, motivating the need for efficient surrogate models that approximate the true utility function while preserving key structural properties like submodularity and monotonicity.

When the true utility function is not monotone submodular, optimizing it directly can be computationally challenging or infeasible. In such cases, it is beneficial to approximate the function with a monotone submodular surrogate. Recent work has shown that it is often possible to design or learn surrogate utility functions that are easier to evaluate and align well with the true objective Ross et al. (2013); Azimi et al. (2012); Das & Kempe (2018). These surrogates can then be used in greedy subset selection algorithms, enabling practical and near-optimal decision-making. However, existing approaches often require hand-crafted surrogates, rely on heuristic assumptions, or do not explicitly enforce submodular structure, limiting their ability to predict performance and reason over unseen elements or contexts.

In this work, we propose a representation-driven learning framework for contextual set functions that generalizes to unseen elements/contexts and supports efficient subset selection. Divergent from the existing work, we consider a broader class of set utility functions that are context-dependent. For example, in a sensor selection task, the utility of a sensor is influenced by environmental conditions such as terrain, weather, or ambient light. Similarly, in a recommendation task, the utility of an item depends on the preferences and characteristics of the specific user to whom it is being recommended. Our key insight is that learning contrastive element embeddings which capture performance similarity conditioned on context can enable generalization to unseen contexts, and the learnt surrogate function, which approximates the submodularity property over the set of elements, can be used for efficient subset selection and re-optimization using greedy algorithms which enjoy near-optimality guarantees.

**Overview of main contributions.** We introduce SELECT - a SEt function LEarning framework via Contrastive elemenT representations. It is a two-stage learning framework consisting of two components: (i) an element representation network that produces context-aware embeddings of set elements, and (ii) a set function approximation network that models the underlying contextual set function.

- First, we propose a contrastive learning framework using the triplet margin loss objective to learn element-wise contrastive embeddings $\phi(x, z)$ that map elements $x$ and context $z$ to a latent space based on singleton utilities obtained from an oracle. We introduce a spectral regularization into this loss function, which aims to prevent dimensional collapse of element embeddings and promotes diverse and rich representations. Under certain smoothing assumptions, we establish that this regularized loss function guarantees asymptotic convergence to a stationary point under projected gradient descent.

- Second, we propose a Deep Sets style architecture Zaheer et al. (2017) to learn (i.e., approximate) the contextual utility function using aggregated element embeddings $\hat{f}(A, z) = \rho \left( \sum_{x \in A} \phi(x, z) \right)$, where $A$ denotes a set and $z$ denotes a context. To ensure that the resulting (surrogate) contextual set function exhibits monotone submodular behavior, we introduce a regularization loss inspired by the LEASURE framework Alieva et al. (2020) for the set-level aggregator network $\rho$. This loss operates on synthetically constructed set quadruples to softly enforce submodularity and monotonicity.

- We evaluate our framework on a wide variety of real-world tasks, including content recommendation (using MovieLens dataset), document summarization (using Reutuers Corpus), and sensor selection in urban environments. For the sensor selection task, we use a high-fidelity simulator to gather data that captures realistic environmental variability of sensor coverage utility. Our results show that the SELECT framework effectively predicts utilities for unseen sets and contexts, consistently outperforming standard baselines in both prediction accuracy and subset selection quality.

## 2 Related Work

**Submodular optimization and greedy selection.** Submodular functions have been widely used in machine learning and decision-making, with applications ranging from document summarization, feature selection, experimental design, and influence maximization Leskovec et al. (2007); Bhargav et al. (2023); Lin & Bilmes (2011); Krause & Golovin (2014). The maximization of submodular functions has been studied extensively under various settings including cardinality-constrained Bhargav et al. (2024), streaming Badanidiyuru et al. (2014), and distributed environments Mirzasoleiman et al. (2015). Greedy algorithms are a core tool in this setting and are known to provide provable approximation guarantees for monotone submodular maximization Nemhauser et al. (1978).

**Learning submodular functions.** Several works have studied learning submodular functions from data. Early work focused on learning mixtures of known submodular templates Yue & Joachims (2008); El-Arini et al. (2009), modular approximations Sipos et al. (2012), or mixtures of concave-over-modular functions Lin & Bilmes (2012). Expressive function classes such as deep submodular functions Dolhansky & Bilmes (2016) allow modeling non-linear interactions between elements while preserving submodularity; however, they are not capable of modeling all submodular functions and do not support contextual utility prediction. In a theoretical setting, the PAC learnability of general submodular functions has been studied in Balcan & Harvey (2011). In contrast to these approaches, our work uses soft regularization to encourage approximate submodularity. Since many real-world utility functions often deviate from strict submodularity, hard enforcement may result in poor approximation. Soft regularization allows the model to learn approximately submodular functions, striking a balance between flexibility and adherence to submodularity.

**Learning set functions for subset selection.** Recent works have studied neural set function approximation, including EquiVSet Ou et al. (2022), which learns neural set functions using an optimal subset oracle, and HORSE Xie et al. (2024), which uses hierarchical attention-based representations for large-scale subset selection. The setting in Ou et al. (2022) assumes an oracle that gives the optimal subset for a given ground set (and not the true utility), whereas our framework assumes that the oracle provides actual utility values for a given subset. Moreover, while attention-based architectures are expressive and useful for modeling interactions Xie et al. (2024), they can introduce substantial computational overhead due to it's quadratic complexity as the ground set grows. Several recent works have explored learning surrogate utility functions to replace expensive evaluation functions in subset selection problems Alieva et al. (2020); Azimi et al. (2012); Kim & Boukouvala (2020). These approaches allow for efficient selection when the true objective is expensive or inaccessible. The LeaSuRe framework Alieva et al. (2020) proposes a principled submodular regularization loss that encourages learned functions to exhibit diminishing returns. Our work builds on this foundation by integrating LeaSuRe-style regularization with a contrastively learned embedding framework, enabling efficient prediction over unseen sets and contexts. Furthermore, our proposed learning algorithm captures contextual dependency of the utility function, outperforming the vanilla LeaSuRe when the utility function varies with different contexts.

**Contrastive learning for structured prediction.** Contrastive learning has emerged as a powerful approach for learning representations by pulling similar instances together and pushing dissimilar ones apart Oord et al. (2018); Chen et al. (2020). It has been successfully applied in domains such as graph learning You et al. (2020), metric learning Schroff et al. (2015), and recommendation systems Liu et al. (2021). In our work, we build on existing contrastive learning frameworks to train context-conditioned element embeddings based on utility-informed similarity. By ensuring that elements with similar singleton utilities under a given context are embedded closer in the representation space, we create a utility-aware latent structure. We show that this helps in the generalization of our model to estimate utilities for unseen subsets and supports rapid re-optimization in dynamic settings.

## 3 Background and Problem Formulation

Let $\mathcal{V}$ be a finite ground set of elements (e.g., sensors, movies), and $\mathcal{Z}$ denote a finite set of context vectors (e.g., environmental conditions, user metadata), such that each context $z \in \mathbb{R}^p$ in the set $\mathcal{Z}$ encodes contextual information. We define a *contextual set function* as $f : 2^{\mathcal{V}} \times \mathcal{Z} \to \mathbb{R}_{\geq 0}$. In many applications, $f$ exhibits *monotonicity* and *submodularity* over sets $S \subseteq \mathcal{V}$. Below, we formalize a new notion of contextual monotonicity and submodularity.

**Definition 1** (Contextual Monotonicity)**.** *A contextual set function $f : 2^{\mathcal{V}} \times \mathcal{Z} \to \mathbb{R}_{\geq 0}$ is monotonically increasing (or non-decreasing) if for every $z \in \mathcal{Z}$ and $S_1 \subseteq S_2 \subseteq \mathcal{V}$ we have $f(S_1, z) \leq f(S_2, z)$.*

**Definition 2** (Contextual Submodularity)**.** *A set function $f : 2^{\mathcal{V}} \times \mathcal{Z} \to \mathbb{R}_{\geq 0}$ exhibits contextual submodularity if it satisfies the following for every $z \in \mathcal{Z}$:*

$$f(S_1 \cup \{x\}, z) - f(S_1, z) \geq f(S_2 \cup \{x\}, z) - f(S_2, z), \quad \forall S_1 \subseteq S_2 \subseteq \mathcal{V}, x \notin S_2.$$

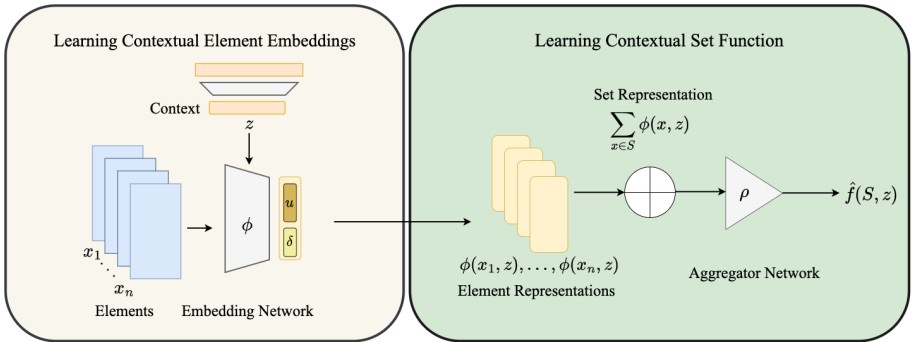

Figure 1: Architecture diagram of the SELECT framework: The first part learns contextual embeddings for elements based on their features and the given context. The second part aggregates these embeddings to represent a set and uses an aggregation network to predict contextual set utility.

Our main goal is to learn a (surrogate) contextual set function $\hat{f}(S, z)$ by querying an oracle for the utilities $f(S, z)$ for set-context pairs $(S, z)$. We wish to learn the surrogate by querying the oracle efficiently, such that it can generalize to unseen subsets and contexts, and maintains a monotone submodular structure. Additionally, since a set function by definition does not consider the order of elements, we want the surrogate to satisfy permutation invariance Zaheer et al. (2017), as formalized below for a contextual set function.

**Definition 3** (Permutation Invariance). *A contextual set function $f : 2^{\mathcal{V}} \times \mathcal{Z} \rightarrow \mathbb{R}_{\geq 0}$ is permutation invariant to the order of elements in the set, i.e., for every context $z \in \mathcal{Z}$ and any permutation $\pi : [M] \rightarrow [M], f(\{x_1, \ldots, x_M\}, z) = f(\{x_{\pi(1)}, \ldots, x_{\pi(M)}\}, z), x_i \in \mathcal{V}$.*

To this end, we propose a representation-driven learning framework in which the set function is modeled via a Deep Sets architecture Zaheer et al. (2017):

$$\hat{f}(S, z) = \rho \left( \sum_{x \in S} \phi(x, z) \right), \tag{1}$$

where $\phi(x, z)$ is contextual element representations, and $\rho$ is an aggregator network.

We split the learning problem into two parts: learning the element embedding network $\phi$ using contrastive learning, and learning the aggregator network $\rho$ using monotonicity and submodularity regularization over set-level utility supervision. These are discussed in the next two sections. The complete SELECT framework is shown in Figure 1.

## 4 Contrastive Learning of Contextual Element Embeddings

We begin by learning a contextual element embedding function $\phi : \mathcal{V} \times \mathcal{Z} \rightarrow \mathbb{R}^d$, where $\mathcal{V}$ is the finite ground set of elements (e.g., movies, sensors) and $\mathcal{Z}$ is the set of context variables (e.g., user information, environmental conditions). Each element $x \in \mathbb{R}^m$ in the ground set $\mathcal{V}$ is described by a potentially large $m$ dimensional feature vector, which can make direct processing computationally expensive. We aim to encode this feature under a given context $z \in \mathcal{Z}$ in a shared latent space which is a compact, low-dimensional representation of high-dimensional elements tailored to specific contexts. The contrastive learning framework ensures elements with similar utility are embedded close together, while dissimilar elements are pushed farther apart. In particular, the geometry of the embedding space should reflect similarity in utility contribution, conditioned on the context.

To achieve this, we employ a contrastive learning objective based on triplet margin loss Jing et al. (2021). Each training example/sample consists of a quadruple $(x_a, x_p, x_n, z)$, where $x_a$ is an anchor element, $x_p$ is a positive element (with similar utility to $x_a$), and $x_n$ is a negative element (with dissimilar utility), all under the same context $z$. The contrastive training dataset $\mathcal{T}$ is constructed using singleton utility feedback. We denote $\mathcal{Z}_{\text{train}} \subseteq \mathcal{Z}$ to be the set of contexts used for training the network. For each context $z \in \mathcal{Z}_{\text{train}}$, we

form a triplet $(x_a, x_p, x_n)$ as follows: we sample an element $(x_a)$ uniformly at random from $\mathcal{V}$. We query the utility oracle for $f(\{x_a\}, z)$. Then, we sample another element $x \in \mathcal{V} \setminus \{x_a\}$ uniformly at random and denote it by $x_p$ (*positive element*) if $|f(\{x_a\}, z) - f(\{x\}, z)| \leq \epsilon$, or denote it by $x_n$ (*negative element*) if $|f(\{x_a\}, z) - f(\{x\}, z)| > \epsilon$, where $\epsilon > 0$ is a similarity threshold, which is a hyperparameter. We repeat this process until a predefined number of samples are gathered or a sampling budget is exhausted. For a context $z$, if no valid triplets $(x_a, x_p, x_n)$ are found, we change $\epsilon$ or extend the sampling pool to ensure sufficient diversity, and repeat the process. Let $\mathcal{T}_z$ denote the set of triplets $(x_a, x_p, x_n)$ for a context $z$, from which we obtain the full training set $\mathcal{T} = \bigcup_{z \in \mathcal{Z}_{\text{train}}} \mathcal{T}_z$.

We employ the triplet margin loss for training the embedding network Khosla et al. (2020). This loss encourages the embedding of $x_a$ to be closer to that of $x_p$ than to that of $x_n$ by at least a margin $\alpha$:

$$\mathcal{L}_{\text{contrastive}}(\phi; \mathcal{T}) = \sum_{(x_a, x_p, x_n, z) \in \mathcal{T}} \max\left(0, \text{sim}(\phi(x_a, z), \phi(x_n, z)) - \text{sim}(\phi(x_a, z), \phi(x_p, z)) + \alpha\right), \qquad (2)$$

where $\text{sim}(\cdot, \cdot)$ denotes a similarity metric over the vector space, such as cosine similarity, and $\alpha > 0$ is a hyperparameter controlling the required margin between positive and negative pairs. This contrastive objective enables the model to learn a representation space in which element embeddings encode contextual utility similarity.

However, training the embedding network $\phi$ only with triplet loss can lead to a phenomenon known as *dimensional collapse* or *rank collapse*, where all inputs are mapped to a small subspace in the embedding space, severely limiting the model's ability to distinguish among different elements Jing et al. (2021) and the expressiveness of the learned set function. This issue has been widely observed in the contrastive learning literature, especially when the number of negatives is small or when regularization is weak Hua et al. (2021); Jiang et al. (2023). Many approaches have addressed this by introducing variance- or decorrelation-based regularization terms that promote feature diversity across dimensions. Notable methods include Barlow Twins Zbontar et al. (2021) and VICReg Bardes et al. (2022). A noteable method SimCLR Chen et al. (2020) relies on contrastive learning with large batches of negative samples and the InfoNCE loss, which encourages instance discrimination. Self-supervised learning frameworks such as BYOL Grill et al. (2020) and SimSiam Chen & He (2021) learn representations by aligning multiple views of the same input without relying on negative samples. However, both approaches rely on architectural tricks, e.g., the use of projection heads, momentum encoders or predictor networks, to avoid representational collapse. In contrast, our framework introduces an explicit regularization-based mechanism to ensure full-rank embeddings, which promotes diversity and informativeness of the learned representations.

Specifically, we split the output vector of the embedding network $\phi(x, z) \in \mathbb{R}^d$ into two components: $u(x, z) \in \mathbb{R}^k$, which is the contrastive component used in the triplet loss, and $\delta(x, z) \in \mathbb{R}^{d-k}$: a complementary component used to increase the diversity and volume of the full embedding space. Here $k$ is a hyperparameter which controls the sizes of the two components. The full embedding is given by

$$\phi(x, z) = \begin{bmatrix} u(x, z) \\ \delta(x, z) \end{bmatrix}, \qquad (3)$$

where the stacked vector representation reflects concatenation. We now denote the modified contrastive loss by $\mathcal{L}_{\text{triplet}}$, which is applied only on $u(x, z)$, given by

$$\mathcal{L}_{\text{triplet}}(\phi; \mathcal{T}) = \sum_{(x_a, x_p, x_n, z) \in \mathcal{T}} \max\left(0, \text{sim}(u(x_a, z), u(x_n, z)) - \text{sim}(u(x_a, z), u(x_p, z)) + \alpha\right). \qquad (4)$$

We note that $\mathcal{L}_{\text{triplet}}$ is piecewise differentiable and sub-differentiable almost everywhere Khosla et al. (2020). In practice, we rely on automatic differentiation frameworks to compute gradients. A smooth surrogate such as a Huberized version of the triplet loss can be substituted for improved gradient flow Xu et al. (2016). We discuss this in detail in Section 4.1.

Next, we apply a independence regularization to the full embedding $\phi(x, z)$, encouraging the entire representation to span a high-volume region in feature space. Recall that $\mathcal{Z}_{\text{train}}$ is the set of contexts used for

training. For each $z \in \mathcal{Z}_{\text{train}}$, define the embedding matrix $U^{(z)} = [\phi(x_1, z), \ldots, \phi(x_n, z)] \in \mathbb{R}^{d \times n}$, where $\{x_1, \ldots, x_n\} = \mathcal{V}$ is the ground set of elements. To encourage the embeddings to be informative and linearly independent under each context, we introduce an independence regularization term:

$$\mathcal{L}_{\text{indep}}(\phi) = \frac{1}{|\mathcal{Z}_{\text{train}}|} \sum_{z \in \mathcal{Z}_{\text{train}}} -\log \det \left( U^{(z)\top} U^{(z)} + \varepsilon I \right), \tag{5}$$

where $\varepsilon I$ is a diagonal matrix with a small positive constant $\varepsilon$ for numerical stability. This loss encourages the Gram matrix of the embeddings under each context to be full-rank, which geometrically corresponds to maximizing the volume of the parallelepiped spanned by the embedding vectors. This promotes the embedding network to map each element to a sufficiently distinct representation in the latent space, facilitating unique representations at the set level. It is worth noting that for large ground sets, the log-determinant regularizer in Equation 5 can be computed using mini-batch estimates or sampled Nyström/anchor-based Gram matrix approximations, which are scalable approaches for large matrices Drineas et al. (2005).

The final contrastive loss used for training the embedding network combines the modified triplet margin loss with the independence regularization:

$$\mathcal{L}_\phi = \mathcal{L}_{\text{triplet}}(\phi; \mathcal{T}) + \lambda_{\text{indep}} \mathcal{L}_{\text{indep}}(\phi), \tag{6}$$

where $\lambda_{\text{indep}}$ is a tunable hyperparameter that controls the strength of the regularization term. The entire procedure is outlined in Algorithm 1.

---

**Algorithm 1** Contrastive Learning of Contextual Element Embeddings

---

**Require:** Element set $\mathcal{V}$, context set $\mathcal{Z}$, number of samples $N$, utility oracle $f$, hyperparameters $\epsilon, \alpha, k$, regularization parameter $\lambda_{\text{indep}}$, epochs $T$, learning rate $\eta$
1: Initialize parameters $\theta_\phi^0$ for embedding network $\phi$
2: **Generate Triplet Dataset $\mathcal{T}$:**
3: **while** $|\mathcal{T}| < N$ **do**
4:     Sample an anchor $x_a \in \mathcal{V}$ and $z \in \mathcal{Z}_{\text{train}}$ uniformly at random
5:     Identify $x_p \in \mathcal{V} \setminus x_a$ s.t. $|f(\{x_a\}, z) - f(\{x_p\}, z)| \le \epsilon$ via uniform random sampling
6:     Identify $x_n \in \mathcal{V} \setminus x_a$ s.t. $|f(\{x_a\}, z) - f(\{x_n\}, z)| > \epsilon$ via uniform random sampling
7:     Add triplet   $\mathcal{T} \leftarrow \mathcal{T} \cup (x_a, x_p, x_n, z)$
8: **end while**
9: **Train $\phi$ network:**
10: **for**  $t = 1$ to $T$ **do**
11:     Obtain embedding vectors $\phi(x_i, z), i = 1, .., n, \forall z \in \mathcal{Z}_{\text{train}}$, using the $\phi$ network
12:     Compute triplet loss $\mathcal{L}_{\text{triplet}}^t$ (as in equation 4) for samples in $\mathcal{T}$ using specfied $\alpha$ and $k$
13:     Compute independence loss $\mathcal{L}_{\text{indep}}^t$ (as in equation 5) using embeddings $\phi(x_i, z)_{i=1}^n \forall z \in \mathcal{Z}_{\text{train}}$
14:     Update network parameters $\theta_\phi^{t+1} \leftarrow \theta_\phi^t - \eta \nabla \mathcal{L}_\phi^t$, where $\mathcal{L}_\phi^t = \mathcal{L}_{\text{triplet}}^t + \lambda_{\text{indep}} \mathcal{L}_{\text{indep}}^t$
15: **end for**
16: **return**  Trained embedding network $\phi$

---

### 4.1 Theoretical Analysis

We now present our theoretical analysis of convergence for the contrastive learning framework under projected gradient descent. We begin by formally stating the assumptions under which our results hold, followed by a few key lemmas establishing smoothness and boundedness properties of the regularized loss function. Building on these lemmas, we present the main convergence result, showing that projected gradient descent on the regularized loss asymptotically converges to a stationary point. We defer all detailed proofs to Appendix A.

**Assumption 1** (Bounded Embeddings and Compact Parameter Space)**.** *For each context $z \in \mathcal{Z}_{\text{train}}$, the embedding matrix $U^z \in \mathbb{R}^{n \times d}$ is uniformly bounded: $\|U^z(\theta_\phi)\|_2 \le R$, where $R \in \mathbb{R}_{>0}$ is some finite constant, and $\theta_\phi \in \Theta$ are the network parameters. Additionally, the parameter space $\Theta \subset \mathbb{R}^p$ is closed and compact.*

**Assumption 2** (Network Regularity). *The embedding network $\phi$ is continuously differentiable ($C^1$). Define $J^z(\theta_\phi) \triangleq \nabla_{\theta_\phi}\phi(x; z)$ as the Jacobian of the embedding output with respect to the network parameters $\theta_\phi$. We have $\|J^z(\theta_\phi)\| \leq K$ and $\|J^z(\theta_\phi) - J^z(\theta'_\phi)\| \leq \rho_J\|\theta_\phi - \theta'_\phi\|$ for all $\theta_\phi, \theta'_\phi \in \Theta$, where $\Theta$ is the compact parameter space, and $K, \rho_J \in \mathbb{R}$ are finite constants.*

Assumptions 1 and 2 are commonly made in convergence analysis of gradient-based optimization methods and can be typically satisfied by projecting parameters onto a compact set, normalizing the embedding vectors and using Lipschitz activation functions (e.g., tanh) in the network Du et al. (2019).

We note that the objective/loss function (as in equation 6) consists of two parts: a triplet-margin loss and a log det regularization term. The triplet margin loss $\mathcal{L}_{\text{triplet}}$ (as in equation 4) suffers from analytical shortcomings: the hinge function $\max(0, \cdot)$ is not differentiable at zero, and its gradient is not Lipschitz continuous (Xu et al., 2016). This is an undesirable property for gradient-based optimization, as smoothness of the loss is a key requirement for convergence. To address this, we adopt a smoothed version of the triplet loss by replacing the hinge with a $C^1$ (continuously differentiable) Huberized hinge $h_\mu(\cdot)$, controlled by a smoothing parameter $\mu > 0$. Let $\zeta(x_a, x_p, x_n) = \text{sim}(u(x_a, z), u(x_n, z)) - \text{sim}(u(x_a, z), u(x_p, z)) + \alpha$. The Huberized triplet-margin loss per triplet is defined as

$$h_\mu(\zeta) = \begin{cases} 0, & \zeta \leq 0, \\ \frac{\zeta^2}{2\mu}, & 0 < \zeta < \mu, , \\ \zeta - \frac{\mu}{2}, & \zeta \geq \mu, \end{cases} \tag{7}$$

where $\mu > 0$ controls the smoothing width. This is similar to the Huberized Support Vector Machine (HSVM) proposed in (Xu et al., 2016), which uses a differentiable approximation of the hinge loss function for SVMs. The resulting smoothed triplet loss is

$$\mathcal{L}^\mu_{\text{triplet}}(\phi, \mathcal{T}) = \sum_{(x_a, x_p, x_n) \in \mathcal{T}} h_\mu(\zeta(x_a, x_p, x_n)). \tag{8}$$

When the violation $\zeta$ is small ($< \mu$), the penalty is quadratic, giving smooth gradients. For large violations, the penalty grows linearly, as in the standard hinge. At the transition points 0 and $\mu$, the function is continuously differentiable. Setting $\mu = 0$ recovers the standard hinge, i.e., $\max(0, \cdot)$, while larger $\mu > 0$ yields smoother behavior. The Huberized hinge function is $\mathcal{O}(1/\mu)$ gradient-Lipchitz Xu et al. (2016).

We employ a time-varying regularization parameter $\varepsilon_t$ in equation 5 and decay it over time to balance numerical stability with representational flexibility. The overall loss function with the Huberized triplet loss as in equation 8 and the time-varying regularization parameter $\varepsilon_t$ is given by

$$\mathcal{L}_{\phi,t} = \mathcal{L}^\mu_{\text{triplet}}(\phi, \mathcal{T}) + \lambda_{\text{indep}} \frac{1}{|\mathcal{Z}_{\text{train}}|} \sum_{z \in \mathcal{Z}_{\text{train}}} -\log\det(U^{(z)\top}U^{(z)} + \varepsilon_t I). \tag{9}$$

**Lemma 1.** *Let Assumptions 1 and 2 hold. Let $\nabla_{\theta_\phi}\mathcal{L}^\mu_{\text{triplet}}(\theta_\phi)$ denote the gradient of the Huberized triplet-margin loss function (as in equation 8) with respect to the network parameters $\theta_\phi$. For all $\theta_\phi, \theta'_\phi \in \Theta$, we have*

$$\left\| \nabla_{\theta_\phi}\mathcal{L}^\mu_{\text{triplet}}(\theta_\phi) - \nabla_{\theta_\phi}\mathcal{L}^\mu_{\text{triplet}}(\theta'_\phi) \right\| \leq L^{\theta_\phi}_{\text{trip}}\|\theta_\phi - \theta'_\phi\|, \tag{10}$$

*where*

$$L^{\theta_\phi}_{\text{trip}} = \mathcal{O}\left( \rho_J |\mathcal{Z}_{\text{train}}|\sqrt{|\bar{\mathcal{T}}_z|} + \frac{K^2}{\mu}|\mathcal{Z}_{\text{train}}||\bar{\mathcal{T}}_z| \right), \quad |\bar{\mathcal{T}}_z| = \max_{z \in \mathcal{Z}_{\text{train}}} |\mathcal{T}_z|. \tag{11}$$

**Lemma 2.** *Let Assumptions 1 and 2 hold. Let $g_z(\theta_\phi) = -\lambda_{\text{indep}}\log\det\left(U^z(\theta_\phi)^\top U^z(\theta_\phi) + \varepsilon I\right)$ and let $\nabla_{\theta_\phi}g_z(\theta_\phi)$ denote the gradient of $g_z(\theta_\phi)$ with respect to the network parameters $\theta_\phi$. For all $\theta_\phi, \theta'_\phi \in \Theta$, we have*

$$\|\nabla_{\theta_\phi}g_z(\theta_\phi) - \nabla_{\theta_\phi}g_z(\theta'_\phi)\| \leq L^{\theta_\phi}_{\text{logdet}}\|\theta_\phi - \theta'_\phi\|, \tag{12}$$

*where*

$$L_{\text{logdet}}^{\theta_\phi} = \frac{2\lambda_{\text{indep}}\rho_J}{\varepsilon}\sqrt{r}\,R \;+\; K^2\left(\frac{2\lambda_{\text{indep}}}{\varepsilon} + \frac{4\lambda_{\text{indep}}R^2}{\varepsilon^2}\right), \quad r = \min\{n,d\}. \tag{13}$$

**Lemma 3.** *Let Assumptions 1 and 2 hold. Let $\nabla_{\theta_\phi}\mathcal{L}_{\phi,t}(\theta_\phi)$ denote the gradient of the loss function (as in equation 9) with respect to the network parameters $\theta_\phi$. From Lemma 1 and Lemma 2, we have, for all $\theta_\phi, \theta'_\phi \in \Theta$,*

$$\left\|\nabla_{\theta_\phi}\mathcal{L}_{\phi,t}(\theta_\phi) - \nabla_{\theta_\phi}\mathcal{L}_{\phi,t}(\theta'_\phi)\right\| \;\leq\; \alpha_t\,\|\theta_\phi - \theta'_\phi\|,$$

*where*

$$\alpha_t \;=\; L_{\text{trip}}^{\theta_\phi} \;+\; \frac{2\,\lambda_{\text{indep}}\,\rho_J\,\sqrt{r}\,R}{\varepsilon_t} \;+\; \frac{2\,\lambda_{\text{indep}}\,K^2}{\varepsilon_t} \;+\; \frac{4\,\lambda_{\text{indep}}\,K^2\,R^2}{\varepsilon_t^2}, \qquad r = \min\{n,d\}.$$

*In other words, the loss function $\mathcal{L}_{\phi,t}$ is $\alpha_t-$smooth (i.e., the gradient of $\mathcal{L}_{\phi,t}$ is $\alpha_t-$Lipchitz) with $\alpha_t = \mathcal{O}\left(\frac{1}{\varepsilon_t^2}\right)$.*

**Lemma 4.** *The loss function $\mathcal{L}_{\phi,t}$ (as in equation 9) has the following properties:*

*(a) Bounded drift:*

$$\left|\mathcal{L}_{t+1,\theta_\phi} - \mathcal{L}_{t,\theta_\phi}\right| \;\leq\; \lambda_{\text{indep}}\,r\,\frac{|\varepsilon_{t+1} - \varepsilon_t|}{\min\{\varepsilon_{t+1}, \varepsilon_t\}}. \tag{14}$$

*(b) Lower bound:*

$$\mathcal{L}_{t,\theta_\phi} \;\geq\; -\,\lambda_{\text{indep}}\,r\,\log(R^2) \quad \forall\,t. \tag{15}$$

Building on Lemma 3 and Lemma 4, we now present the main convergence result for projected gradient descent applied to the time-varying loss function defined in equation 9.

**Theorem 1.** *Let Assumptions 1 and 2 hold. Consider the projected gradient descent (PGD) updates on the time varying loss function in equation 9 given by*

$$\theta_{\phi,t+1} = \text{Proj}_\Theta\big(\theta_{\phi,t} - \eta_t\nabla_{\theta_\phi}\mathcal{L}_t(\theta_{\phi,t})\big), \tag{16}$$

*where $\nabla_{\theta_\phi}\mathcal{L}_t(\theta_{\phi,t})$ is the gradient of the loss function with respect to $\theta_\phi$. Let $\varepsilon_t = \varepsilon_0(t+\kappa)^{-\beta}$ with $0 < \beta < \frac{1}{2}$ and $\kappa > 1$, so that $\alpha_t = \mathcal{O}((t+\kappa)^{2\beta})$ and $\eta_t = \gamma/\alpha_t$ with $\gamma \in (0,1)$. Then*

$$\lim_{t\to\infty}\|\nabla\mathcal{L}_t(\theta_t)\| = 0. \tag{17}$$

*In other words, PGD on the time-varying loss function in equation 9 with a decaying regularization parameter $\varepsilon_t = \mathcal{O}((t+\kappa)^{-\beta}), 0 < \beta < \frac{1}{2}, \kappa > 1$, and learning rate $\eta_t = \gamma/\alpha_t$ converges to a stationary point, asymptotically.*

Now, we have the following result that follows from Theorem 1.

**Corollary 1.** *By the asymptotic convergence in Theorem 1, there exists a limit parameter $\theta_\infty \in \Theta$ such that, for every context $z \in \mathcal{Z}_{\text{train}}$, the embedding matrix $U^z(\theta_\infty)$ is full-rank, given $d \geq n$. As a result, the set representation $\sum_{x\in S}\phi_{\theta_\infty}(x,z)$ is injective. In other words, for each context $z$, and for any two distinct subsets $X, Y \subseteq \mathcal{V}, X \neq Y$,*

$$\sum_{x\in X}\phi_{\theta_\infty}(x,z) \;\neq\; \sum_{y\in Y}\phi_{\theta_\infty}(y,z). \tag{18}$$

Since our framework uses a sum-based aggregator whose output feeds into the function approximation network ($\rho$) (see Section 5), injectivity of the set representation (as stated in Corollary 1) is essential as different sets must map to distinct embeddings to ensure that the approximation network $\rho$ can differentiate between them.

**Remark 1.** *When $d < n$, injective set representations can still be achieved by assigning a dedicated coordinate in the $\delta$ subspace of the embedding of each element with unique positive values, guaranteeing uniqueness of subset sums. This construction simultaneously preserves semantic closeness in the $u$ subspace, which continues to capture utility-based similarity among elements. While it is true that full linear independence cannot be enforced when $d < n$, this does not diminish the relevance of the log-determinant regularizer. Its role is not to*

*impose strict independence, but to actively encourage geometric diversity by penalizing collapsed or low-rank configurations of the embeddings. This promotes spread and volume in the representation space, which is critical for stable learning and discriminative structure. In contrast, when $d \geq n$, the embedding space has sufficient degrees of freedom to naturally support linear independence and injectivity, allowing these properties to emerge as a consequence of optimization rather than through explicit structural enforcement.*

## 5  Learning the Surrogate Utility Function

Once the element embeddings $\phi(x, z)$ are learned, we turn our attention to learning the set-level aggregator function $\rho : \mathbb{R}^d \to \mathbb{R}$, which predicts the utility of a set given the sum of the embeddings of its elements. We construct a training dataset $\mathcal{D}_{\text{real}} = \{(S_i, z_i, y_i)\}_{i=1}^N$ consisting of set-context-utility triples. This is done by sampling set-context pairs $(S_i, z_i) \in 2^{\mathcal{V}} \times \mathcal{Z}_{\text{train}}$ uniformly at random and $y_i = f(S_i, z_i)$ is obtained by querying the utility oracle. The empirical loss for this supervised learning task is defined as:

$$\mathcal{L}_{\text{mse}}(\hat{f}, f) = \sum_{(S,z,y) \in \mathcal{D}_{\text{real}}} (\hat{f}(S, z) - y)^2. \tag{19}$$

However, simply fitting $\mathcal{L}_{\text{mse}}$ does not guarantee that the learned model is submodular. If the true function $f$ is submodular, then this approach may capture its behavior. However, if $f$ is only approximately submodular or not submodular at all, then minimizing $\mathcal{L}_{\text{mse}}$ will fit the network to whatever structure $f$ has, and the learned model may not exhibit submodular/ approximate submodular structure. To address this, we need explicit regularization terms that can, in principle, bias the learning process toward submodularity and ensure that the model $\rho$ approximates any $f$ by its *closest approximately submodular function*.

We construct a synthetic regularization dataset $\mathcal{D}_{\text{synth}}$ (i.e., the utility oracle will not be queried/evaluated for creating this data) comprising of quadruples $(X, Y, W, Z)$. For each $S_i, z \in \mathcal{D}_{\text{real}}$, we select sets $X, Y$ uniformly at random such that $X \subset Y \subseteq S_i$, and let $W = X \cup \{e\}, Z = Y \cup \{e\}$ for some element $e \in \mathcal{V} \setminus Y$, also sampled uniformly at random. These quadruples are designed to expose potential violations of the diminishing returns property, which is a defining characteristic of submodular functions.

To encourage submodularity in the learned surrogate $\hat{f}(\cdot, z)$, we define the submodularity regularization loss:

$$\mathcal{L}_{\text{sub}}(\hat{f}, f) = \sum_{(X,Y,W,Z) \in \mathcal{D}_{\text{synth}}} \left[ \text{ReLU}\left( \hat{f}(Z, z) - \hat{f}(Y, z) - (\hat{f}(W, z) - \hat{f}(X, z)) \right) \right], \tag{20}$$

which penalizes deviations from the submodular property, i.e., $\hat{f}(Z, z) - \hat{f}(Y, z) \leq \hat{f}(W, z) - \hat{f}(X, z)$. In particular, the ReLU function ensures that the loss is non-negative and only activated when the submodularity inequality is violated. If the inequality holds (i.e., the marginal gain from adding $e$ to the larger set $Y$ is no greater than adding it to the smaller set $X$), the loss contributes zero. This selective activation makes the loss both efficient to compute and tightly focused on penalizing only true violations, without unnecessarily affecting valid examples Alieva et al. (2020). To further ensure that the learned surrogate exhibits monotonicity, we define a monotonicity regularization term:

$$\mathcal{L}_{\text{mono}}(\hat{f}, f) = \sum_{(X,W) \in \mathcal{D}_{\text{synth}}} \left[ \text{ReLU}(\hat{f}(X, z) - \hat{f}(W, z)) \right], \tag{21}$$

which penalizes cases where adding an element decreases the predicted utility. Again, ReLU activation ensures that only violations of monotonicity contribute to the loss, maintaining the non-negativity of the penalty. As discussed in Alieva et al. (2020), these synthetic regularization terms shape the learned function $\hat{f}$ to approximate monotone submodular behavior while requiring supervision only on small subsets. Our final training objective combines these components:

$$\mathcal{L}_{\rho}(\hat{f}, f) = \mathcal{L}_{\text{mse}}(\hat{f}, f) + \lambda_{\text{sub}} \mathcal{L}_{\text{sub}}(\hat{f}, f) + \lambda_{\text{mono}} \mathcal{L}_{\text{mono}}(\hat{f}, f), \tag{22}$$

where $\lambda_{\text{sub}}$ and $\lambda_{\text{mono}}$ are regularization parameters for the submodularity and monotonicity regularizations, respectively. We outline the surrogate learning procedure in Algorithm 2.

---

**Algorithm 2** Learning Surrogate Utility Function $\hat{f}$

---

**Require:** Embeddings $\phi(x, z)$ for $x, z \in \mathcal{V} \times \mathcal{Z}$, dataset $\mathcal{D}_{\text{real}} = \{(S_i, z_i, f(S_i, z_i))\}_{i=1}^N$, regularization parameters $\lambda_{\text{sub}}, \lambda_{\text{mono}}$, epochs $T$, learning rate $\eta$, number of regularization samples $N_s$

1: Initialize parameters $\theta_\rho^0$ for aggregator network $\rho$
2: **Construct Synthetic Quadruples Dataset** $\mathcal{D}_{\text{synth}}$:
3: **for** each $S_i \in \mathcal{D}_{\text{real}}$ until $|\mathcal{D}_{\text{synth}}| = N_s$ **do**
4:     Sample subsets $X \subset Y \subseteq S_i$ and element $e \notin Y$, uniformly at random
5:     Form quadruples $(X, Y, W = X \cup \{e\}, Z = Y \cup \{e\})$
6:     $\mathcal{D}_{\text{synth}} \leftarrow \mathcal{D}_{\text{synth}} \cup (X, Y, W, Z)$
7: **end for**
8: **Train $\rho$ network**:
9: **for** $t = 1$ to $T$ **do**
10:    Estimate loss function $\mathcal{L}_\rho^t(\hat{f}, f) = \mathcal{L}_{\text{mse}}^t(\hat{f}, f) + \lambda_{\text{sub}} \mathcal{L}_{\text{sub}}^t(\hat{f}, f) + \lambda_{\text{mono}} \mathcal{L}_{\text{mono}}^t(\hat{f}, f)$ using $\mathcal{D}_{\text{real}}$ & $\mathcal{D}_{\text{synth}}$
11:    Update network parameters: $\theta_\rho^{t+1} \leftarrow \theta_\rho^t - \eta \nabla \mathcal{L}_\rho^t$
12: **end for**
13: **return** Trained surrogate network $\rho$

---

## 6 Experiments

We evaluate our framework through a comprehensive set of experiments on content recommendation, sensor selection, and document summarization tasks designed to answer the following key questions: (i) Does the representation learning framework in SELECT effectively cluster similar elements while being able to discriminate between sets of elements?, (ii) Can the SELECT framework predict utilities of unseen sets and contextual variables accurately?, and (iii) Can the surrogate utility model learned using the SELECT framework drive effective downstream subset selection?

**Baselines and Comparisons.** We evaluate SELECT for its ability to predict utilities for unseen sets by varying the train to test split against the following state-of-the-art baselines: (i) Vanilla Deep Sets Zaheer et al. (2017), trained directly on set-utility pairs without monotonicity and submodularity regularization; (ii) LEASURE Alieva et al. (2020), where the set elements are represented as a one-hot encoding vector (as opposed to embeddings), but have the monotonicity and submodularity regularization terms. In Appendix B, we present additional details regarding datasets, model architectures, and training. We also present additional experiments with noisy oracle feedback, re-optimization under shifts, and runtime comparisons with baselines.

### 6.1 Content Recommendation (MovieLens Dataset).

We evaluate our framework on a movie recommendation task using the MovieLens dataset Harper & Konstan (2015). In this setup, each user is treated as a contextual variable $z \in \mathbb{R}^4$, described via demographic features such as age, gender, occupation, and zip code, while each movie is treated as a ground-set element $x$, encoded using its genre-based binary feature vector. For each user $z$, the utility of a set of movies $X \subseteq \mathcal{V}$ is $f(X, z) = \max_{x \in X} \text{rating}(x, z)/5$, which captures the user's preference for their most highly-rated movie in the set. This form models the set utility using a max-coverage-style objective aligned with user satisfaction, which is monotone-submodular Chen et al. (2017). We train SELECT by sampling a subset of 10 users (as contexts $\mathcal{Z}$) and 50 movies (as the ground set $\mathcal{V}$) uniformly at random along with their corresponding utility labels. This allows us to evaluate the ability of our framework to learn from user-movie interactions and test generalization to unseen movie combinations and users.

**Representation Quality:** We analyze the singular value spectrum of the matrix formed by stacking the embeddings $\phi(x, z)$ across elements and contexts. Specifically, we plot the logarithm of the singular values with and without the log-determinant regularization term in Figure 2a. Larger singular values correspond to a higher effective rank, indicating a richer and more diverse feature space. Our results show that log-determinant regularization significantly improves the spread of singular values, resulting in a higher-rank embedding matrix. This suggests that the regularization encourages the embeddings to span more volume.

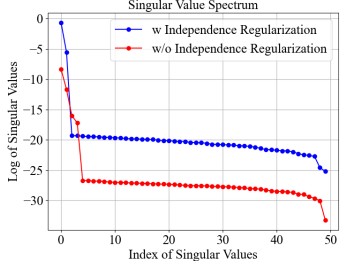

(a) Singular Value Spectrum of Embedding Space

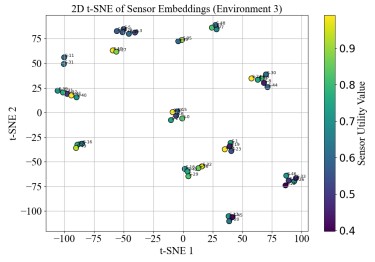

(b) Clustering of Element Embeddings based on Context and Utility

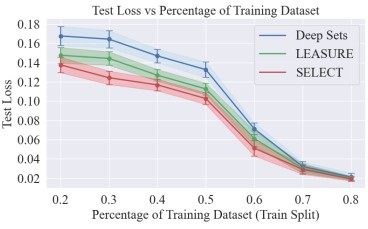

(c) Comparision of SELECT with Deep Sets and LEASURE for Movie Recommendation Task

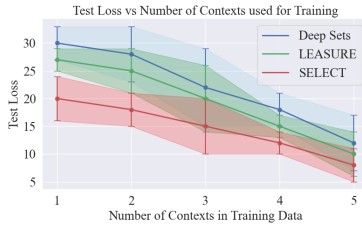

(d) Comparision of SELECT with Deep Sets and LEASURE for Sensor Utility Prediction

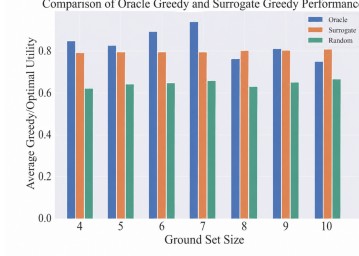

(e) Performance of SELECT in Greedy Sensor Selection

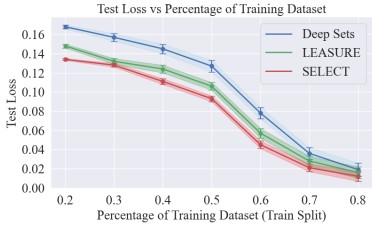

(f) Comparision of SELECT with Deep Sets and LEASURE for Document Summarization Task

Figure 2: Empirical Evaluation of the SELECT Framework

**Embedding Visualization.** We apply t-SNE (t-Distributed Stochastic Neighbor Embedding van der Maaten & Hinton (2008)) technique to visualize the $d-$ dimensional element embeddings $\phi(x, z)$ in 2-D (see Figure 2b). The visualization reveals clustering patterns among items, which reflect the model's ability to encode interactions between elements and contexts based on performance similarity.

**Comparison with Baselines.** We compare SELECT with Vanilla Deep Sets and LEASURE and plot the test loss with varying train-test splits in Figure 2c. We observe that SELECT outperforms both Deep Sets and LEASURE frameworks, with the lowest test loss. In other words, SELECT can more accurately predict the utility of unseen sets compared to Deep Sets and LEASURE frameworks.

### 6.2 Sensor Selection in Urban Environments

**Dataset Generation and Experimental Setup.** We consider a sensor selection task where the objective is to learn the utilities of sets of camera sensors $S$ that is defined by a coverage map of way-points under varying environmental conditions (captured by the context vector $z$). We note that the area coverage utility function $f(A, z)$ is monotone submodular over sets $A$ Lin & Bilmes (2012). We use a high-fidelity photorealistic simulator in a drone detection setup built on Unreal Engine 5 as our utility oracle. In our experiments, we consider a ground set $\mathcal{V}$ of 4 camera sensors with varied characteristics. Each sensor is encoded in the feature vector $x$ using its intrinsic parameters (e.g., FOV, range, resolution, frame rate), while each environment is encoded into vector $z$ using meta-information such as visibility and light levels. To obtain the coverage utility of different sensor sets under varying environmental conditions, we run a YOLO-based detection pipeline Diwan et al. (2023) under distinct weather conditions (e.g., clear sky, low-light cloudy, heavy fog, etc.) for detecting drones and obtained the coverage map (i.e., number of way-points covered by sensors).

We provide further details of the experimental setup in Appendix B. We apply the proposed SELECT framework to learn contextual coverage utility of sensor sets. Using singleton feedback from drone detection coverage maps, we train a context-conditioned contrastive embedding network $\phi(x, z)$ by setting $\epsilon = 0.1$. We train the set-level surrogate $\hat{f}(A, z) = \rho\left(\sum_{x \in A} \phi(x, z)\right)$, using the supervision dataset $\mathcal{D}_{\text{real}}$, which consists of subsets $A \subseteq \mathcal{V}$ of varying sizes, with labels obtained from the simulator $y_{A,z} = f(A, z)$.

**Comparison with Baselines.** We compare the contextual transfer ability of SELECT with Deep Sets and LEASURE by progressively increasing the number of contexts (i.e., environment conditions) it is trained on $\mathcal{Z}_{\text{train}}$ and plot the average test loss over all sensor sets and contexts in $\mathcal{Z}_{\text{test}}$ (i.e., new environment conditions) in Figure 2d. We observe that SELECT outperforms Deep Sets and LEASURE, with lower prediction loss under the contextual shift. This shows that our framework can effectively predict utilities of sensors in new/unseen environmental conditions.

**Downstream Subset Selection using Greedy.** To evaluate the effectiveness of the learned surrogate utility function $\hat{f}$ in downstream tasks, we employ it as an oracle within a greedy selection framework and compare its performance against greedy selection using the true utility oracle. Specifically, we compute the ratio of the achieved utility to the optimal utility for both the surrogate-based and true-oracle-based greedy selections. This evaluation is conducted over varying ground set sizes. For each size $n$, we run greedy selection with subset budgets ranging from 1 to $n-1$ and plot the average performance across these instances in Figure 2e. The results demonstrate that the surrogate function, when used with greedy selection, achieves utility values that are closely aligned with those obtained using the true utility oracle and clearly outperforms those chosen randomly. This highlights the effectiveness of the surrogate model learning the true utility, even when trained on only 50% of subsets.

### 6.3 Personalized Document Summarization

We evaluate our framework on a personalized document summarization task based on the Reuters Corpus Lewis et al. (2004). This can be seen as a contextual subset selection problem, where one has to generate a personalized summary of articles for users based on their topic importance or preference. We apply Latent Dirichlet Allocation (LDA) to the Reuters Corpus with $n_{\text{topic}} = 10$ topics. Here, the elements $x \in \mathcal{V}$ are articles and the context $z$ corresponds to the user's topic preference vector. The utility $f(S, z)$ models the personalized coverage of set $S$ for user $z$. Each article $x \in \mathcal{X}$ is associated with a topic distribution $P(i|x)$ over topics $i \in \{1, \ldots, n_{\text{topic}}\}$. A *user* is characterized by a *preference vector* $z \in \Delta^{n_{\text{topic}}}$ (a L1 normalized weight vector), representing their affinity to different topics. We define a *contextual utility function* $f(S, z)$ over a set of articles $S$ and user preference $z$ as follows: $f(S, z) = \sum_{i=1}^{n_{\text{topic}}} z_i \left(1 - \prod_{x \in S}(1 - P(i|x))\right)$. This defines a *context-dependent coverage score* quantifying how well the set $S$ covers the user's topical interests, weighted by their preferences. We note that this utility function is monotone submodular Chen et al. (2017). Our framework is trained on multiple such $(S, z)$ pairs and evaluated on held-out users and subsets. Note that we do not aggregate utilities over users in our framework (as in Chen et al. (2017)), as our goal is to generalize to unseen contexts (i.e., new user preference vectors), not just average performance. Similar to our previous experiments, we vary the proportion of training subsets and evaluate the test loss on held-out subsets. Again, we compare the performance of SELECT against Deep Sets and LEASURE, and observe that SELECT consistently achieves lower test loss across varying training set sizes (see Figure 2f).

## 7 Conclusion

We proposed a framework for learning surrogates for contextual set functions with monotone submodular structure. By combining contrastive learning of element embeddings with submodular regularization on the set-level representations, our approach enables generalization to unseen sets and contexts using limited supervision samples from the true utility oracle. We present a theoretical analysis of convergence for the regularized contrastive learning framework and show that it leads to diverse element embeddings, promoting richer set representations. We show that combining this with the synthetic submodular-norm regularization, which softly enforces structure in the surrogate model, supports efficient greedy optimization. Empirical studies on content recommendation, document summarization, and sensor selection tasks demonstrate that our method generalizes effectively across contexts and outperforms baselines in utility prediction and decision quality. This framework offers a principled and scalable approach for learning surrogate set functions for structured, context-dependent decision-making problems, with limited supervision samples. Future work will focus on extending the framework to support addition of new set elements or contextual variables without requiring complete retraining or expensive fine-tuning and exploring an end-to-end learning (joint learning of representation and function approximation networks) variant of the framework with theoretical guarantees.

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

# A  Theoretical Proofs

## A.1  Proof of Lemma 1

*Proof.* We first present the bound for a fixed context $z$ and a corresponding triplet $(x_a, x_p, x_n) \in \mathcal{T}_z$. Let $u_a(\theta_\phi) = \phi_{\theta_\phi}(x_a, z)$, $u_p(\theta_\phi) = \phi_{\theta_\phi}(x_p, z)$, $u_n(\theta_\phi) = \phi_{\theta_\phi}(x_n, z)$. As a shorthand, we drop the dependence on $\theta_\phi$ throughout the proof, e.g., $u_a$ instead of $u_a(\theta_\phi)$. Let the Huberized triplet loss per triplet, per context, be defined as

$$\ell^\mu_{\text{triplet}}(\theta_\phi) \;=\; h_\mu\Big(\zeta(\theta_\phi)\Big), \quad \zeta(\theta_\phi) = \cos(u_a, u_n) - \cos(u_a, u_p) + \alpha,$$

where $h_\mu$ is as defined in equation 7. From the analysis presented in Xu et al. (2016), we have $h_\mu \in C^1$, $|h'_\mu(\zeta)| \le 1$ for all $\zeta$, and $h'_\mu$ is $(1/\mu)$-Lipschitz (i.e., $|h'_\mu(\zeta) - h'_\mu(\zeta')| \le \frac{1}{\mu}|\zeta - \zeta'|$).

The cosine similarity between two unit norm vectors $u, v$, given by $\cos(u, v) = u^\top v$, is 1-Lipschitz. We have the following bounds for $\cos(\cdot)$.

$$\| \cos(u, v) - \cos(u', v') \| \le \|u - u'\| + \|v - v'\|, \|\nabla_u \cos(u, v)\| \le 1, \|\nabla_v \cos(u, v)\| \le 1.$$

These follow from bilinearity and Cauchy–Schwarz inequality. By network regularity (Assumption 2), we have $\|J(\theta_\phi)\| \le K$, $\|J(\theta_\phi) - J(\theta'_\phi)\| \le \rho \|\theta_\phi - \theta'_\phi\|$. By the mean-value theorem, we have $\|\phi_{\theta_\phi}(x, z) - \phi_{\theta'_\phi}(x, z)\| \le K\|\theta_\phi - \theta'_\phi\|$.

Now, we apply the chain rule to evaluate the gradient of the Huberized triplet loss function,

$$\nabla_{\theta_\phi} \ell^\mu_{\text{triplet}}(\theta_\phi) = h'_\mu(\zeta(\theta_\phi)) \, \nabla_{\theta_\phi} \zeta(\theta_\phi).$$

Applying the network regularity bound on the gradient of $\zeta$, we get

$$\|\nabla_{\theta_\phi} \zeta(\theta_\phi)\| \le \|J(\theta_\phi)\| \, \|u_n - u_p\| + \|J(\theta_\phi)\| \, \|u_a\| + \|J(\theta_\phi)\| \, \|u_a\| \le 4K.$$

Now, we consider the norm-difference of the gradients, given by

$$\|\nabla_{\theta_\phi} \ell^\mu_{\text{triplet}}(\theta_\phi) - \nabla_{\theta_\phi} \ell^\mu_{\text{triplet}}(\theta'_\phi)\| \le |h'_\mu(\zeta(\theta_\phi)) - h'_\mu(\zeta(\theta'_\phi))| \, \|\nabla \zeta(\theta_\phi)\| + |h'_\mu(\zeta(\theta'_\phi))| \, \|\nabla\zeta(\theta_\phi) - \nabla\zeta(\theta'_\phi)\|.$$

Now we have

$$|\zeta(\theta_\phi) - \zeta(\theta'_\phi)| \le 4K \, \|\theta_\phi - \theta'_\phi\| \quad \Rightarrow \quad |h'_\mu(\zeta(\theta_\phi)) - h'_\mu(\zeta(\theta'_\phi))| \le \frac{4K}{\mu} \, \|\theta_\phi - \theta'_\phi\|.$$

With $\|\nabla\zeta(\theta_\phi)\| \le 4K$, this part contributes $\dfrac{16K^2}{\mu} \|\theta_\phi - \theta'_\phi\|$.

By directly applying term-by-term bounds, we have

$$\|\nabla\zeta(\theta_\phi) - \nabla\zeta(\theta'_\phi)\| \le (4\rho_J + 4K^2) \, \|\theta_\phi - \theta'_\phi\|.$$

Combining the above bounds, we have

$$\|\nabla_{\theta_\phi} \ell^\mu(\theta_\phi) - \nabla_{\theta_\phi} \ell^\mu(\theta'_\phi)\| \le \left( \tfrac{16K^2}{\mu} + 4\rho_J + 4K^2 \right) \|\theta_\phi - \theta'_\phi\|.$$

Now, we will sum over these bounds for all triplets for a context $z$ in the training data. For $\mathcal{L}^\mu_{\text{triplet}, z} = \sum_{(x_a, x_p, x_n) \in \mathcal{T}_z} \ell^\mu$, the *size* of the gradient across triplets typically accumulates in Frobenius norm because each triplet selects only three rows of the embedding matrix (Petersen et al. (2008))

$$\|\nabla_{\theta_\phi} \mathcal{L}^\mu_{\text{triplet}, z}(\theta_\phi)\| \;\le\; G^\phi_z \quad \text{with} \quad G^\phi_z = O(\sqrt{|\mathcal{T}_z|}).$$

Next, the *Lipschitz constant* across the sum adds the per-triplet constants, giving

$$\|\nabla_{\theta_\phi} \mathcal{L}^\mu_{\text{triplet},z}(\theta_\phi) - \nabla_{\theta_\phi} \mathcal{L}^\mu_{\text{triplet},z}(\theta'_\phi)\| \leq L^\phi_z \|\theta_\phi - \theta'_\phi\|, \qquad L^\phi_z = O\Big(\frac{|\mathcal{T}_z|}{\mu}\Big)$$

Combining the above terms, we get

$$\big\|\nabla_{\theta_\phi} \mathcal{L}^\mu_{\text{triplet},z}(\theta_\phi) - \nabla_{\theta_\phi} \mathcal{L}^\mu_{\text{triplet},z}(\theta'_\phi)\big\| \leq \Big(\rho_J\, G^\phi_z + K^2 L^\phi_z\Big) \|\theta_\phi - \theta'_\phi\|,$$

with $G^\phi_z = O(\sqrt{|\mathcal{T}_z|})$ and $L^\phi_z = O(|\mathcal{T}_z|/\mu)$ as stated above.

Without loss of generality, we replace $|\mathcal{T}_z|$ with $|\bar{\mathcal{T}}_z| = \max_{z \in \mathcal{Z}_{\text{train}}} |\mathcal{T}_z|$. Finally, multiplying by number of contexts in the training data $|\mathcal{Z}_{\text{train}}|$, we obtain the stated bound. □

## A.2 Proof of Lemma 2

*Proof.* In this proof, we use standard results from Matrix Calculus for Positive Definite matrices which can be found in Petersen et al. (2008). We use the following shorthands throughout the proof: $\lambda = \lambda_{\text{indep}}$, $U = U^z(\theta_\phi)$ and $V = U^z(\theta'_\phi)$.

We have $g_z(\theta_\phi) = -\lambda_{\text{indep}} \log\det\big(U^z(\theta_\phi)^\top U^z(\theta_\phi) + \varepsilon I\big)$. For $\varepsilon > 0$, we have that $U^z(\theta_\phi)^\top U^z(\theta_\phi) + \varepsilon I \succ 0$, i.e., is Positive Definite. By applying chain rule and standard matrix calculus Petersen et al. (2008), the derivative of this log det term is given by

$$\nabla_{\theta_\phi} g_z(\theta_\phi) = J^z(\theta_\phi)^\top \big[ -2\lambda\, U^z(\theta_\phi)\, (U^z(\theta_\phi)^\top U^z(\theta_\phi) + \varepsilon I)^{-1}\big],$$

where $J^z(\theta_\phi)$ is the Jacobian of $U^z(\theta_\phi)$ w.r.t. $\theta_\phi$.

For any $U$, $\|\nabla g(U)\|_F \leq \frac{2\lambda}{\varepsilon} \|U\|_F$. Now, using $\|U^z(\theta_\phi)\|_F \leq \sqrt{r}R$, we obtain $\|\nabla_{\theta_\phi} g_z(\theta_\phi)\| \leq \frac{2\lambda K}{\varepsilon} \sqrt{r}\, R$.

For any $\theta_\phi, \theta'_\phi \in \Theta$, we have

$$\|\nabla_{\theta_\phi} g_z(\theta_\phi) - \nabla_{\theta_\phi} g_z(\theta'_\phi)\| \leq \|J^z(\theta_\phi) - J^z(\theta'_\phi)\|\, \|\nabla g(U^z(\theta_\phi))\|_F$$
$$+ \|J^z(\theta'_\phi)\|\, \|\nabla g(U^z(\theta_\phi)) - \nabla g(U^z(\theta'_\phi))\|_F.$$

The first term is bounded by

$$\rho_J \|\theta_\phi - \theta'_\phi\| \cdot \frac{2\lambda}{\varepsilon} \sqrt{r}\, R.$$

.

To bound the second term, we use the following standard results directly (see Petersen et al. (2008); Boyd & Vandenberghe (2004)) for $A = U^\top U + \varepsilon I$ : $\|A^{-1}\| \leq 1/\varepsilon$ and $A^{-1} - B^{-1} = A^{-1}(B - A)B^{-1}$, where $B = V^\top V + \varepsilon I$. Combining the two bounds, we obtain

$$\|\nabla_U g(U) - \nabla_U g(V)\|_F \leq \Big(\frac{2\lambda}{\varepsilon} + \frac{4\lambda R^2}{\varepsilon^2}\Big) \|U - V\|_F \quad \text{whenever } \|U\|_2, \|V\|_2 \leq R.$$

We have $\|U^z(\theta_\phi) - U^z(\theta'_\phi)\|_F \leq K\|\theta_\phi - \theta'_\phi\|$. Thus, the second term is bounded by

$$K^2 \Big(\tfrac{2\lambda}{\varepsilon} + \tfrac{4\lambda R^2}{\varepsilon^2}\Big) \|\theta_\phi - \theta'_\phi\|.$$

Adding both bounds yields the stated Lipschitz constant $L^{\theta_\phi}_{\text{logdet}}$, given by

$$L^{\theta_\phi}_{\text{logdet}} = \frac{2\lambda_{\text{indep}}\rho_J}{\varepsilon} \sqrt{r}\, R \; + \; K^2 \Big(\frac{2\lambda_{\text{indep}}}{\varepsilon} + \frac{4\lambda_{\text{indep}} R^2}{\varepsilon^2}\Big), \quad r = \min\{n, d\}.$$

□

### A.3 Proof of Lemma 3

*Proof.* This follows directly by applying triangle inequality and then combining the bounds from Lemma 1 and Lemma 2. □

### A.4 Proof of Lemma 4

*Proof.* (a) In this drift bound, we evaluate the maximum difference between the loss functions under the same network parameters $\theta_\phi$, but between two time steps $t$ and $t+1$. The first terms remains unchanged since it only depends on the network parameters. It now remains to analyze the difference in the second term. Let $A_z = U^{z\top}U^z \succeq 0$ and define $h_z(\varepsilon) = -\log\det(A_z + \varepsilon I)$. By the mean value theorem,

$$|h_z(\varepsilon_{t+1}) - h_z(\varepsilon_t)| = |h_z'(\varepsilon')|\,|\varepsilon_{t+1} - \varepsilon_t|$$

for some $\varepsilon'$ between $\varepsilon_t$ and $\varepsilon_{t+1}$. Since $h_z'(\varepsilon) = -\mathrm{tr}((A_z + \varepsilon I)^{-1}) \leq r_z/\varepsilon$, we obtain

$$|h_z(\varepsilon_{t+1}) - h_z(\varepsilon_t)| \leq \frac{r}{\min\{\varepsilon_{t+1}, \varepsilon_t\}}\,|\varepsilon_{t+1} - \varepsilon_t|,$$

where $r = \min(n, d)$. Averaging over $z$ and multiplying by $\lambda_{\mathrm{indep}}$, we obtain the stated bound.

**Note:** For the schedule $\varepsilon_t = \varepsilon_0(t + \kappa)^{-\beta}$, $|\varepsilon_{t+1} - \varepsilon_t| = \mathcal{O}((t + \kappa)^{-\beta-1})$ and $\min\{\varepsilon_{t+1}, \varepsilon_t\} = \Theta((t + \kappa)^{-\beta})$, so the bound is $\mathcal{O}(1/(t + \kappa)^\beta)$.

(b) The first term, i.e., the Huberized triplet loss, is nonnegative ($\geq 0$). We only need to examine the regularization term to establish a lower bound.

$$\mathcal{L}_{t,\theta_\phi} \geq \lambda_{\mathrm{indep}}\frac{1}{|\mathcal{Z}_{\mathrm{train}}|}\sum_z -\log\det(U^{z\top}U^z + \varepsilon_t I).$$

Since $\|U^z\|_2 \leq R$, we have $U^{z\top}U^z \preceq R^2 I$. Thus

$$-\log\det(U^{z\top}U^z + \varepsilon_t I) \geq -r\log(R^2 + \varepsilon_t) \geq -r\log(R^2).$$

Averaging over $z$, we obtained the stated finite lower bound. □

### A.5 Proof of Theorem 1

*Proof.* The proof is based on a standard argument using the *Descent Lemma* for $\alpha_t$-smooth functions. We first establish a one–step decrease inequality for the time-varying loss, then control the drift induced by the changing $\varepsilon_t$ using the learning rate $\eta_t$, and finally apply a telescoping sum argument. Comparing the growth rates of the descent and drift terms leads to the conclusion that the projected gradient descent iterates converge to a stationary point.

We denote $\mathcal{L}_{t,\theta_{\phi,t}}$ by $\mathcal{L}_t(\theta_{\phi,t})$. The *Descent Lemma* for $\alpha_t$-smooth functions with learning rate $\eta_t$ is given by

$$\mathcal{L}_t(\theta_{\phi,t+1}) \leq \mathcal{L}_t(\theta_{\phi,t}) - \eta_t\left(1 - \frac{\eta_t\alpha_t}{2}\right)\|\nabla\mathcal{L}_t(\theta_{\phi,t})\|^2.$$

By applying the *Projected Gradient Descent* update as in equation 16 with learning rate $\eta_t = \gamma/\alpha_t$,

$$\mathcal{L}_t(\theta_{\phi,t+1}) \leq \mathcal{L}_t(\theta_{\phi,t}) - \frac{c_\gamma}{\alpha_t}\|\nabla\mathcal{L}_t(\theta_{\phi,t})\|^2,$$

where $c_\gamma = \gamma(1 - \gamma/2) > 0$.

From Lemma 4, we have

$$\mathcal{L}_{t+1}(\theta_{\phi,t+1}) \leq \mathcal{L}_t(\theta_{\phi,t+1}) + \frac{C}{t + \kappa},$$

where $C > 0$ is some universal constant. Combining this, we get

$$\mathcal{L}_{t+1}(\theta_{\phi,t+1}) \leq \mathcal{L}_t(\theta_{\phi,t}) - \frac{c_\gamma}{\alpha_t}\|\nabla\mathcal{L}_t(\theta_{\phi,t})\|^2 + \frac{C}{t+\kappa}.$$

Summing from $t = 0$ to $T - 1$ and using $\mathcal{L}_t \geq L_{\min}$ (where $L_{\min}$ is the lower-bound which characterized in Lemma 4 (b)),

$$c_\gamma \sum_{t=0}^{T-1} \frac{1}{\alpha_t}\|\nabla\mathcal{L}_t(\theta_{\phi,t})\|^2 \leq \mathcal{L}_0(\theta_{\phi,0}) - L_{\min} + C\log(T+\kappa). \tag{23}$$

With $\varepsilon_t = \varepsilon_0(t+\kappa)^{-\beta}$, we have $\alpha_t = \Theta((t+\kappa)^{2\beta})$. Therefore

$$\sum_{t=0}^{T-1} \frac{1}{\alpha_t} = \mathcal{O}(T+\kappa)^{1-2\beta}.$$

For $0 < \beta < \frac{1}{2}$, this grows polynomially in $T$, much faster than the $\log(T)$ growth on the right-hand side of equation 23. Now suppose, for contradiction, that there exists $\epsilon > 0$ such that $\|\nabla\mathcal{L}_t(\theta_{\phi,t})\| \geq \epsilon$ for infinitely many $t$. Then the left-hand side of equation 23 would be bounded below by $c_\gamma\epsilon^2 \sum_{t\in I,\,t<T} \frac{1}{\alpha_t}$, where $I$ is the infinite set of indices where the gradient is large. This sum diverges at rate $(T+\kappa)^{1-2\beta}$, contradicting the logarithmic growth on the right-hand side. This means that the gradients cannot remain bounded away from zero infinitely.

In other words, the choice of learning rate $\eta_t$ and $\varepsilon_t$ schedule is such that it balances the growth of $\alpha_t$ in a way that ensures that the asymptotic sum of gradients remains finite.

As a result, we have

$$\lim_{t\to\infty}\|\nabla\mathcal{L}_t(\theta_{\phi,t})\| = 0,$$

which proves convergence to stationary points, asymptotically. $\qquad\square$

## A.6 Proof of Corollary 1

*Proof.* By Theorem 1, the projected gradient descent iterates satisfy $\lim_{t\to\infty}\|\nabla\mathcal{L}_t(\theta_{\phi,t})\| \to 0$, so there exists a limit point $\theta_\infty \in \Theta$. Fix $z \in \mathcal{Z}_{\text{train}}$ and let $A^z(\theta) = U^z(\theta)^\top U^z(\theta) \succeq 0$ with eigenvalues $\{\lambda_i^z(\theta)\}_{i=1}^d$. Then the independence term is

$$-\log\det\big(A^z(\theta) + \varepsilon_t I\big) = -\sum_{i=1}^d \log\big(\lambda_i^z(\theta) + \varepsilon_t\big).$$

Its gradient with respect to $U^z$ is $\nabla_{U^z}\big(-\log\det(A^z + \varepsilon_t I)\big) = -2\,U^z(\theta)\,(A^z(\theta) + \varepsilon_t I)^{-1}$, whose operator norm scales like $\max_i 1/(\lambda_i^z(\theta) + \varepsilon_t)$.

By Theorem 1, we have $\theta_{\phi,t} \to \theta_\infty$ and $\|\nabla\mathcal{L}_t(\theta_{\phi,t})\| \to 0$ while $\varepsilon_t \to 0$. If, for contradiction, some $\lambda_i^z(\theta_\infty) = 0$, then by continuity $\lambda_i^z(\theta_{\phi,t}) + \varepsilon_t \to 0$, which makes $\|(A^z(\theta_{\phi,t}) + \varepsilon_t I)^{-1}\| \to \infty$ and forces the gradient contribution from the log-det term to blow up, contradicting $\|\nabla\mathcal{L}_t(\theta_{\phi,t})\| \to 0$. Hence, for all $i$, $\lambda_i^z(\theta_\infty) > 0$. As a result, we have $U^z(\theta_\infty)$ is full rank for all $z \in \mathcal{Z}_{\text{train}}$.

Full rank of $U^z(\theta_\infty)$ when $d \geq n$ implies linear independence of the embedding vectors $\{\phi_{\theta_\infty}(x,z)\}$; therefore the set-sum map $S \mapsto \sum_{x\in S}\phi_{\theta_\infty}(x,z)$ is injective, i.e., for $X \neq Y$ one has $\sum_{x\in X}\phi_{\theta_\infty}(x,z) \neq \sum_{y\in Y}\phi_{\theta_\infty}(y,z)$.
$\qquad\square$

### A.7 Statistical and Geometric Interpretation of the Log-Determinant Regularizer

In this section, we discuss the statistical and geometric interpretations of the log-determinant regularizer in Equation 5, highlighting its connection to maximum-likelihood covariance estimation, maximum entropy, and volume maximization.

Equation 5 includes the log-determinant regularizer $\log \det(U^{z\top} U^z)$, where $U^z \in \mathbb{R}^{m \times d}$ denotes the matrix whose rows are the embedding vectors produced by the network $\phi$, i.e., $e_i = \phi(x_i, z)$ for $i = 1, \ldots, m$. This regularization term encourages the learned embeddings to span a high-volume representation space and prevents collapse into a low-dimensional subspace.

A statistical interpretation can be obtained through a Gaussian covariance model Chatfield (2018). Suppose the embeddings are viewed as samples from a zero-mean Gaussian distribution,

$$e_i \sim \mathcal{N}(0, \Sigma).$$

The log-likelihood of $m$ samples is

$$\ell(\Sigma) = -\frac{m}{2} \log \det(\Sigma) - \frac{1}{2} \sum_{i=1}^{m} e_i^\top \Sigma^{-1} e_i + \text{constant}.$$

The maximum-likelihood estimate of the covariance is the empirical covariance

$$\widehat{\Sigma} = \frac{1}{m} U^\top U.$$

Evaluating the likelihood at this estimate yields a term involving $-\log \det(U^\top U)$ up to additive constants and scaling factors. Thus, optimizing the log-determinant regularizer can be interpreted as encouraging a high-volume covariance structure for the embedding distribution, consistent with a maximum-likelihood covariance interpretation.

The same term is also connected to maximum entropy. For a Gaussian random vector, we have the entropy given by

$$H(\mathcal{N}(0, \Sigma)) \triangleq \frac{1}{2} \log \det(\Sigma) + \text{constant}.$$

Therefore, increasing the log-determinant of the empirical covariance promotes a high-entropy embedding distribution, encouraging the embeddings to occupy diverse directions in the representation space.

Independently of any probabilistic assumption, the regularizer also admits a purely geometric interpretation. Let $\sigma_1, \ldots, \sigma_d$ denote the singular values of $U$. Then

$$\det(U^\top U) = \prod_{j=1}^{d} \sigma_j^2, \qquad \log \det(U^\top U) = 2 \sum_{j=1}^{d} \log \sigma_j.$$

Maximizing this quantity discourages small singular values and prevents spectral collapse. Equivalently, it maximizes the generalized variance and the volume spanned by the learned embeddings. Hence, the log-determinant term admits both a statistical justification and a distribution-free geometric justification, through volume maximization and spectral non-degeneracy.

## B    Additional Experimental Details

For all settings where $d < n$, we assign the last coordinate of the embedding vector $\phi(x, z)$ with a unique integer identifier for the element of ground set to ensure set injectivity (See Remark 1). We set $N_s$ in Algorithm 1 as $N_s = 2|\mathcal{D}_{real}|$.

### B.1 Experimental Setup for MovieLens Recommendation Task

In this section, we provide experimental evaluation details for the content recommendation task using the MovieLens 1M dataset Harper & Konstan (2015) as a benchmark.

### B.1.1   Data Extraction and Preprocessing

The dataset contains user-movie ratings along with user demographic metadata and movie genres.

**User Feature Extraction ($z$):**   Each user is represented by a contextual vector $z \in \mathbb{R}^4$, constructed as follows:

- **Age**: Integer bucket encoded as one of $\{1, 18, 25, 35, 45, 50, 56\}$

- **Gender**: Binary value $\in \{0, 1\}$

- **Occupation**: Integer value between 0–20 (21 unique values)

- **Zip code**: First digit (regional code), converted to integer $\in \{0, \dots, 9\}$

We sample a subset of 10 users uniformly at random to form the context set $\mathcal{Z}$.

**Movie Feature Extraction ($x$):**   Each movie is represented by a binary genre vector $x \in \{0, 1\}^{19}$, indicating whether the movie belongs to each of the 19 possible genres in the dataset. We uniformly sample 50 movies from the dataset to serve as the ground set $\mathcal{V}$.

**Utility Label Generation ($f(X, z)$):**   Given a user $z$ and a subset of movies $X \subseteq \mathcal{V}$, the utility is computed as:
$$f(X, z) = \max_{x \in X} \frac{\text{rating}(x, z)}{5},$$
where $\text{rating}(x, z)$ denotes the user's rating for movie $x$, if available. Ratings are normalized by 5 to lie in $[0, 1]$. Sets with no known ratings are excluded from training.

**Triplet Generation:**   For contrastive embedding training, we construct triplets $(x_a, x_p, x_n)$ for each user $z$ such that:

- $x_a$ is a movie rated above threshold ($> 3.5$)

- $x_p$ is another movie rated similarly high by the same user ($\geq 2.5$)

- $x_n$ is a movie rated significantly lower ($< 2.5$)

These triplets are used to train the embedding network via triplet loss and independence regularization.

**Set Utility Dataset:**   We generate utility values for 1000 subsets $X$ of size 2 to 5 for each user context $z$, using their actual ratings, to train the surrogate submodular model.

### B.1.2   Network Architectures and Hyperparameters

**Training Details:**   The contrastive embedding model is trained first using triplet and independence losses. The resulting embeddings are frozen and fed into the submodular surrogate model, which is trained to minimize MSE along with regularizers enforcing submodularity and monotonicity. The training and evaluation pipeline is repeated across varying train/test splits (e.g., 70/30, 80/20).

### B.1.3   Additional Experiments

We evaluate our framework for its robustness to noisy oracle feedback, its performance on downstream re-optimization tasks, and present runtime comparisons with respect to baselines. We perform the following experiments on the MovieLens dataset.

Table 1: Hyperparameters and Network Architecture for Contrastive Learning of Embeddings ($\phi$)

| Component | Details |
| --- | --- |
| **Embedding Model Architecture** | |
| *Input Dimension* | 23 (19 movie genres + 4 user metadata) |
| *Intermediate Layers* | Dense(32) |
| *Output Dimension* | $d = 8$ (softplus activation) |
| *Embedding Split* | $k = 6$ (utility), $d - k = 2$ (independence) |
| | |
| **Training Hyperparameters** | |
| *Learning Rate* | $10^{-4}$ |
| *Batch Size* | 32 |
| *Number of Epochs* | 50 |
| *Triplet Margin* | $\delta_{\mathrm{margin}} = 0.1$ |
| *Independence Loss Weight* | $\lambda_{\mathrm{indep}} = 0.5$ |
| | |
| **Optimizer** | Adam Optimizer |

Table 2: Hyperparameters and Network Architecture for Surrogate Set Function ($\rho$)

| Component | Details |
| --- | --- |
| **Set Function Model Architecture** | |
| *Input Dimension* | Embedding Size = 8 |
| *Hidden Layers* | Dense(128, activation=ReLU), Dense(64) |
| *Output Layer* | Dense(1, activation=sigmoid) |
| | |
| **Training Hyperparameters** | |
| *Learning Rate* | Initial $10^{-3}$, Exponential Decay |
| *Decay Steps* | 10 |
| *Decay Rate* | 0.2 |
| *Number of Epochs* | 50 |
| *Loss Weights* | $\lambda_{\mathrm{mono}} = 0.1$, $\lambda_{\mathrm{sub}} = 0.5$ |
| | |
| **Optimizer** | Adam Optimizer with Learning Rate Schedule |

**Robustness to Noisy Feedback.** To assess the robustness of our framework against noisy feedback, we inject Gaussian noise $\mathcal{N}(0, \sigma^2)$ into the utility oracle $f$ and evaluate its impact on test performance. We compare SELECT with the Deep Sets framework by varying $\sigma$ incrementally from 0.02 to 0.14 in steps of 0.02. For each noise level, we perform a 70/30 train-test split and measure the test loss over the evaluation dataset.

We plot the test loss vs. noise level in Figure 3a. It clearly demonstrates that SELECT consistently achieves lower test loss across all noise levels compared to Deep Sets. This highlights the effectiveness of our method in maintaining robust generalization performance even under perturbations in the training data.

**Re-optimization under Shifts.** To test the robustness of our learned item embeddings, we simulate partial failure/unavailability scenarios during movie recommendation. We start by selecting 8 movies from a pool of 15 using a greedy algorithm guided by the true oracle. Then, we remove up to 6 highly rated movies from this set.

To recover from these unavailable movies which were selected for recommendation, we compare two strategies:

- **Oracle Re-optimization**, where we re-run the greedy algorithm with the true oracle to pick replacements, and

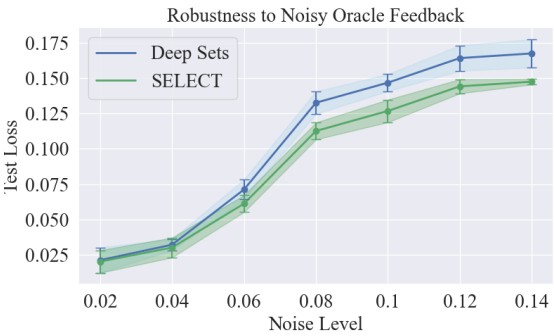
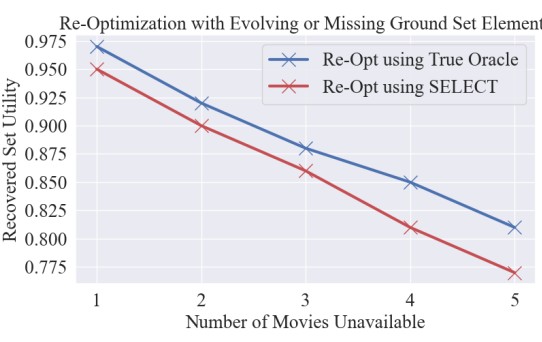

(a) Evaluation of SELECT under Noisy Oracle    (b) Comparision of SELECT for

Figure 3: Evaluation of SELECT for Robustness against Noise and Failures

- **Embedding-based Recovery**, where we replace unavailable movies by selecting the most similar ones (as measured by cosine similarity of their embeddings) using our learned embeddings, without retraining or querying the oracle.

We measure how much utility each recovered set retains compared to the re-optimization using the true oracle. By varying number of unavailable movies, we find that the embedding-based approach preserves a high fraction of the utility and performs close to the oracle re-optimization. This shows that our model supports fast, effective recovery using embedding similarity alone, without having to perform expensive re-optimization from scratch.

**Runtime Comparison.** We report a runtime comparison based on the time required for the training loss to reach a threshold of 0.1 (see Table 3). This provides a consistent criterion for assessing the convergence speed across models.

Table 3: Runtime comparison of SELECT with Deep Sets and LEASURE

| Model | Time to reach training loss $\leq 0.1$ (in min) |
|---|---|
| Deep Sets | $5.5 \pm 0.3$ |
| SELECT | $6.1 \pm 0.1$ |
| LEASURE | $7.2 \pm 0.2$ |

As shown in Table 3, SELECT requires slightly more time to converge compared to Deep Sets, which we attribute to the additional log-determinant regularization. However, this slower convergence comes with the advantage of consistently achieving lower validation losses, highlighting a trade-off between computation time and generalization performance.

## B.2 Ablation Studies

We conduct a series of controlled ablation studies to isolate the contribution of each design choice in SELECT, covering anchor sampling strategy, submodularity approximation quality, hyperparameter sensitivity, the effect of independence regularization on the embedding space, and oracle query efficiency. Unless stated otherwise, all experiments are run on the MovieLens dataset, use the base configuration ($k$=6, $\alpha$=0.1, $\lambda_{\text{indep}}$=1) and are averaged over five independent runs.

### B.2.1 Coreset-Based Anchor Sampling

In Algorithm 1, the anchor element $x_a$ is sampled uniformly from the ground set $\mathcal{V}$. While simple, uniform sampling may systematically under-represent small or sparsely populated regions of the data distribution, biasing the contrastive objective towards high-density clusters and leaving rare elements which often carry the most discriminative information about the utility landscape underexplored.

To address this, we replace uniform sampling with a **coreset-based anchor pool** constructed via *farthest-first traversal* Jubran et al. (2020). Starting from one randomly chosen element, we iteratively add the element that is farthest from the current coreset in feature space as follows.

$$x_t = \arg\max_{x_i \in \mathcal{V}} \min_{x_j \in C_{t-1}} \|x_i - x_j\|_2. \tag{24}$$

We set $|C| = \lfloor \sqrt{|\mathcal{V}|} \rfloor$ and sample anchors from $C$ rather than from $\mathcal{V}$ directly. Farthest-first traversal is a 2-approximation to the $k$-centre problem, guaranteeing that no region of the feature space is more than twice as far from the nearest anchor as the optimal placement would allow Jubran et al. (2020). This ensures that contrastive triplets cover the full diversity of the ground set, so that the learned representation is not dominated by the geometry of dense clusters.

Figure 4 shows that coreset anchoring yields consistently lower test loss across all ground-set sizes, with the gap widening as $|\mathcal{V}|$ grows and diversity becomes harder to capture by chance. The Principal Component Analysis (PCA) projection confirms that coreset anchors are spread uniformly across the embedding space, whereas uniform anchors cluster in dense regions and leave sparse areas unrepresented.

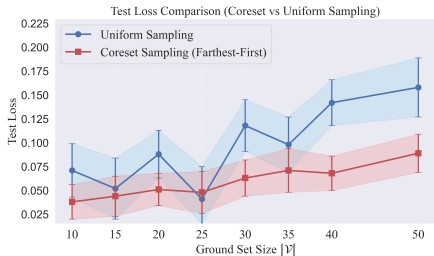
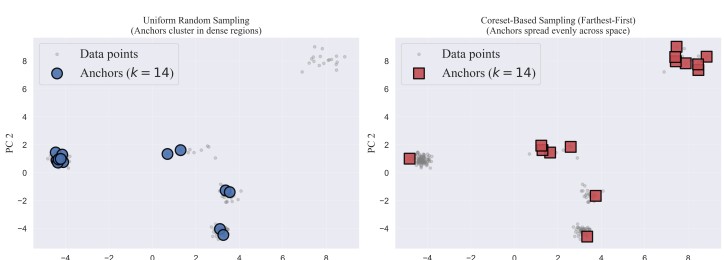

(a) Test loss vs. ground-set size for coreset vs. uniform anchor selection.

(b) 2-D PCA projection of selected anchors. Coreset anchors cover the embedding space more uniformly than uniform samples.

Figure 4: Coreset vs. uniform anchor allocation. Coreset selection achieves lower test loss across all ground-set sizes and better spatial diversity of anchors.

### B.2.2 Submodularity Approximation Quality

A central design goal of SELECT is to learn a surrogate utility function that is not merely accurate on training sets but also approximately submodular that makes greedy maximization tractable and provides near-optimality guarantee. A surrogate that frequently violates the diminishing-returns property will produce greedy solutions that do not inherit this guarantee, degrading downstream performance regardless of its point-wise accuracy. To quantify how well submodularity is preserved, we sample tuples $(A, B, e)$ with $A \subseteq B$ and $e \in \mathcal{V} \setminus B$, and check whether the learned surrogate $\hat{f}$ satisfies

$$\hat{f}(A \cup \{e\}) - \hat{f}(A) \geq \hat{f}(B \cup \{e\}) - \hat{f}(B). \tag{25}$$

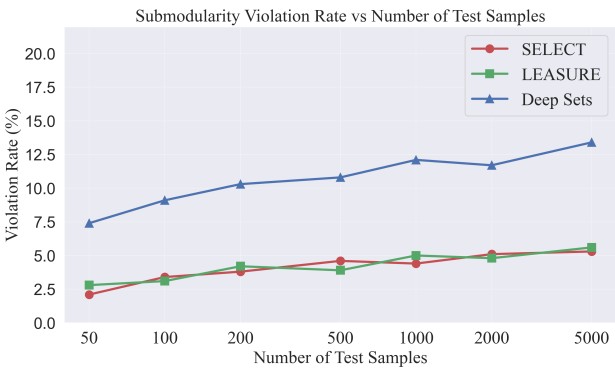

Figure 5: Submodularity violation rate (%) vs. number of test samples.

We report the *violation rate*, i.e., the percentage of sampled tuples that violate this inequality as the number of test samples grows.

As shown in Figure 5, SELECT achieves substantially lower violation rates than LEASURE and Deep Sets across all sample sizes. This confirms that the combination of the contrastive representation and the submodular surrogate loss $\mathcal{L}_{\text{sub}}$ actively promotes diminishing-returns structure in the learned utility, and that this structure is preserved at test time on unseen sets.

### B.2.3 Ablations on Hyperparameters $k$ and $\alpha$

Two key hyperparameters govern the contrastive representation stage: the embedding split $k$ and the triplet margin $\alpha$. We study their effect on test loss to characterise the robustness of the framework and to provide guidance for practitioners.

**Embedding split $k$.** The parameter $k$ partitions the $d = 8$ embedding dimensions into a $k$-dimensional *utility subspace* and a $(d - k)$-dimensional *diversity subspace*. The utility subspace captures how much an element contributes to the set value on its own; the diversity subspace captures how much it complements other elements already in the set. As shown in Figure 6a, test loss is minimised at $k = 6$. When $k$ is too small, the utility signal is under-represented and the model cannot reliably distinguish high-value from low-value elements. When $k$ is too large, the diversity subspace is starved of capacity and the contrastive margin $\alpha$ cannot be satisfied, causing the triplet loss to rise and the learned embeddings to collapse. The least loss occurs at $k = 6$, reflecting the fact that utility variation is the dominant signal in our datasets, but a non-trivial diversity component is still needed for accurate set-level prediction.

**Triplet margin $\alpha$.** The margin $\alpha$ governs the minimum cosine-distance separation enforced between positive and negative pairs in the contrastive stage, controlling the *granularity* of the utility-aware representation. A larger margin pushes embeddings of elements with different utility values further apart, sharpening the representation, but also makes the constraint harder to satisfy with finite data. Figure 6b shows that performance peaks at $\alpha$=0.1. Smaller values provide insufficient separation, causing the embedding space to be under-structured and the aggregator to overfit. Larger values impose overly strict constraints that cannot be reliably satisfied given the available training data, leading to increased triplet loss and degraded generalization.

### B.2.4 Embedding Spectrum

We note that the spectrum in Figure 2a decays quickly due to the low intrinsic dimensionality of the dataset: the movie embeddings are largely captured by a small number of genre directions. To isolate the effect of

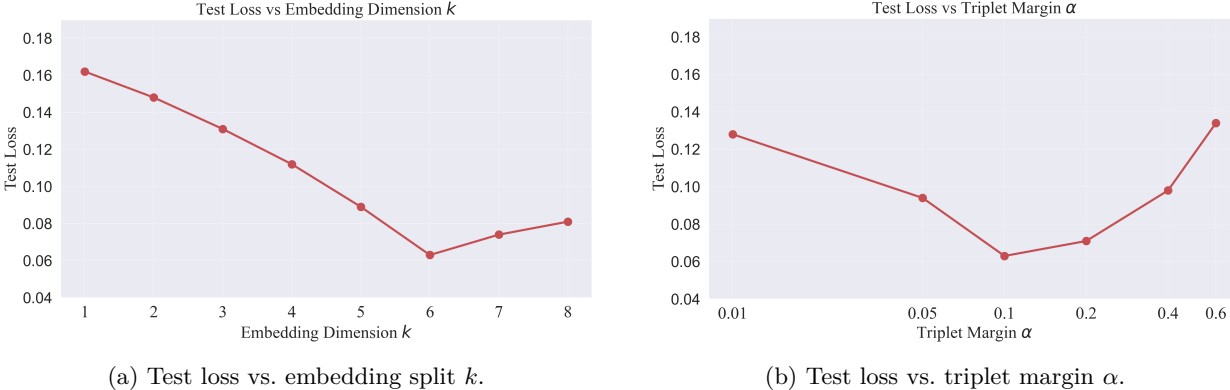

(a) Test loss vs. embedding split $k$.

(b) Test loss vs. triplet margin $\alpha$.

Figure 6: Ablation studies on SELECT for the embedding split $k$ and triplet margin $\alpha$.

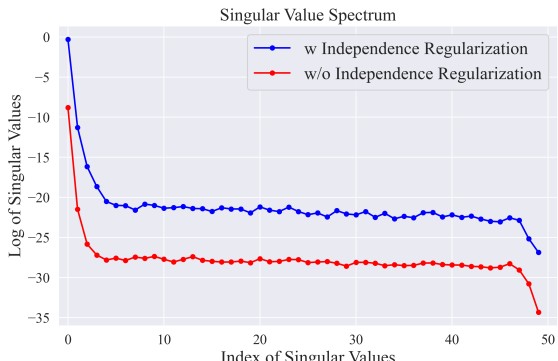

Figure 7: Singular value spectrum of the embedding matrix on a more diverse dataset.

$\mathcal{L}_{\text{indep}}$ from this dataset, we repeat the experiment by sampling a more diverse sample of movies and users. Figure 7 shows that without independence regularization (red), the spectrum remains heavily concentrated in the top components, although compared to the previous case we now observe a few additional eigenvalues of meaningful magnitude, indicating that a few more directions become useful. Beyond those leading components, however, the spectrum still decays quickly. In contrast, with independence regularization (blue), the spectrum is noticeably flatter, with more eigenvalues retaining significant mass across the first few indices. This indicates that the learned representations use a larger number of meaningful directions and therefore capture richer structural information about the ground set.

### B.2.5 Performance Comparison with Budget-Matched Oracle Queries

SELECT uses two types of oracle feedback: $B_{\text{single}}$ singleton utility queries to train the contrastive representation, and $B_{\text{set}}$ set-level utility queries to train the aggregator. A natural question is whether the performance advantage over baselines simply reflects access to a larger total number of labelled samples, rather than a qualitatively better use of the available supervision.

To isolate this, we conduct a *budget-matched* comparison in which all methods receive the same total number of oracle-labelled samples. We fix the ground set to $n=10$ elements and vary the total oracle budget as

$$B_{\text{total}} = n + \frac{2^n - n}{k}, \quad k \in \{5, 4, 3, 2\}, \tag{26}$$

yielding $B_{\text{total}} \in \{212, 263, 348, 517\}$. SELECT uses the split $B_{\text{single}}=n$ singleton queries for the contrastive stage and $B_{\text{set}}=(2^n-n)/k$ set-level queries for the aggregator, while the baselines receive the full $B_{\text{total}}$ budget exclusively for set-level utility learning.

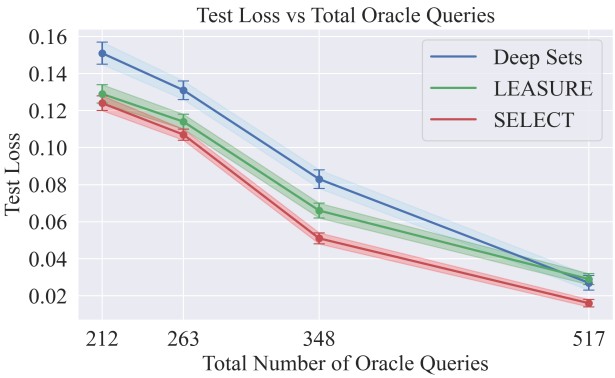

Figure 8: Test loss vs. total oracle queries under the budget-matched oracle queries. SELECT outperforms LEASURE and Deep Sets at every budget level, demonstrating that its gains arise from representation quality rather than additional supervision.

Figure 8 shows that SELECT consistently outperforms both baselines across all budget levels, with the advantage most pronounced at low budgets where the structured representation provides the greatest benefit and the baselines suffer most from data scarcity. This confirms that the performance gains of SELECT stem from the quality of its representations rather than from access to additional supervision.

The key insight is that singleton labels are not only cheaper to obtain in practice, requiring evaluation of a single element rather than a full set, but also more informative per label for learning the representation, since they directly supervise the utility subspace of the embedding without the combinatorial noise introduced by set-level aggregation. The contrastive stage thus converts a small number of singleton labels into a structured, utility-aware representation that improves generalization to unseen sets, a form of *representation leverage* that the baselines cannot exploit regardless of how many set-level labels they receive.

### B.3 Estimating Sensor Utilities in Urban Environments

In this section, we describe the experimental setup for evaluating our framework for a sensor utility prediction and subset selection task in urban environments using data from a high-fidelity photorealistic simulator.

### B.3.1 Simulator Overview

We utilize SYGARD (Synthetic Data Generator for Automatic Target Recognition, Identification, and Detection), a high-fidelity simulation platform developed in Unreal Engine 5, designed to support large-scale image data generation for training and evaluating object detection algorithms. SYGARD integrates the Colosseum API (AirSim) for UAV flight control and Cesium for real-world terrain reconstruction, enabling the simulation of various UAV scenarios within photorealistic urban and rural environments. The platform outputs synchronized RGB imagery, object masks, and ground truth metadata (including location, orientation, and object class) from the camera at configurable frame rates, poses, and image resolutions. The system provides fine-grained control over environmental variables, allowing users to configure weather and lighting conditions to simulate diverse sensing scenarios. This includes the ability to introduce sensor noise effects such as rain, fog, and dust, enabling the generation of data under both ideal and degraded visibility conditions.

This flexible architecture allows researchers to simulate a wide range of operational contexts and generate varied datasets tailored to recognition, detection, and tracking tasks. By automating data collection workflows and standardizing annotation outputs, SYGARD provides a practical tool for generating controlled, high-diversity datasets to support aerial perception research.

### B.3.2 Data Generation and Detection Evaluation

To begin the data generation, we discretize the area of interest into a uniform grid and generate a set of UAV waypoints that collectively span the full region. These waypoints are used to define the UAV's trajectory, ensuring systematic coverage of the urban environment. Once the trajectory is set, fixed ground-based cameras are configured by specifying their field of view, resolution, and pose. Environmental and lighting conditions are then applied using the parameterized weather control system.

During simulation, data is captured at each waypoint the UAV reaches. For every frame, the system records RGB images from each camera, along with the corresponding ground truth metadata, including object class, location, and orientation.

After simulation, the collected metadata is postprocessed to generate bounding box annotations in YOLO format. We then run inference using the *YOLOv11m* model, which was pre-trained on the dataset provided by *Real World Object Detection Dataset for Quadcopter Unmanned Aerial Vehicle Detection*. The model's predictions are compared against the ground truth annotations to assess detection performance.

To visualize and analyze detection outcomes, we map the correctly predicted bounding box centers to spatial coordinates within the scene. This enables evaluation of how well the protected airspace was covered under the given camera and environment configuration. Figure 9 shows the full pipeline from data capture to detection mapping.

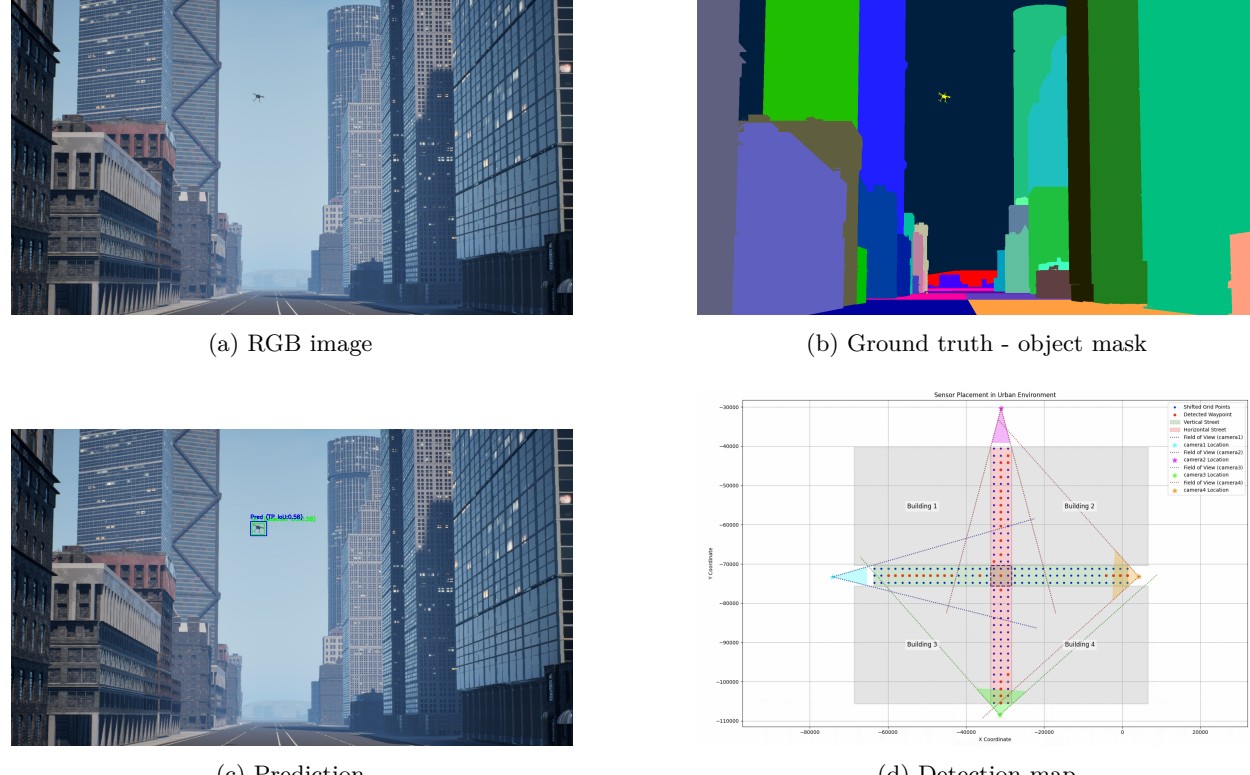

(a) RGB image

(b) Ground truth - object mask

(c) Prediction

(d) Detection map

Figure 9: Detection pipeline visualization: (a) RGB image captured during simulation, (b) generated ground truth object mask used for automatic labeling, (c) prediction from YOLOv11m, and (d) spatial map of correctly detected UAV locations.

### B.3.3 Urban Environment Details

For the experimental setting, we used an urban environment constructed with a third-party plugin — *Brushify Urban Buildings Pack* — in Unreal Engine. This environment includes modular street layouts, 3D

building assets, and infrastructure commonly found in cityscapes, enabling realistic simulation of urban aerial monitoring scenarios. We focus on two orthogonal streets forming a cross-shaped intersection. Each street spans approximately 650 meters in length and 53 meters in width, covering a combined area of roughly $66,332 \, \text{m}^2$, including the overlapping intersection. This configuration is used to study UAV coverage and visibility from fixed ground-based cameras deployed for drone surveillance and detection in urban environments. It provides a testbed for evaluating perception models under realistic occlusions caused by buildings and street geometry.

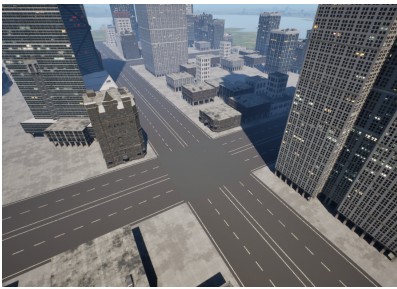

Figure 10: A top-down view of the simulated urban city environment used in our experiments, constructed using the *Brushify Urban Buildings Pack* in Unreal Engine. The environment includes modular streets and building assets suitable for simulating UAV detection scenarios.

### B.3.4 Camera Configuration and Coverage Strategy

Four synchronized camera sensors are deployed to maximize coverage of the intersection area:

- **Long-range cameras** (30° field of view), positioned at opposite corners of the intersection to provide high-resolution distant views.

- **Short-range (wide-angle) cameras** (90° field of view), mounted closer to the center of the intersection for broad coverage and minimal blind spots.

The cameras capture full-HD (1920×1080) or QHD (2560×1440) imagery, with mounting positions and orientations manually specified in the simulation configuration. To evaluate the impact of sensor layout on detection performance, we permute the camera placements across trials—interchanging long-range and short-range sensor positions to produce alternate coverage configurations.

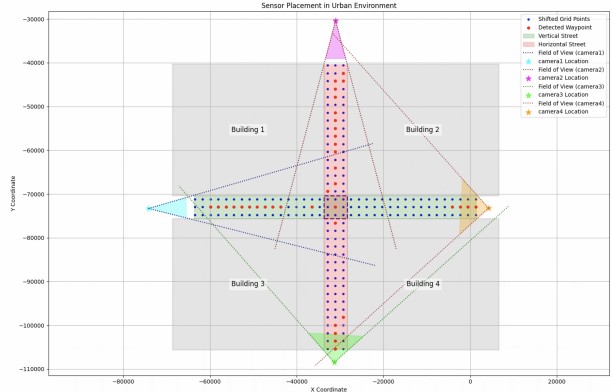

Figure 11: Camera placement and field of view (FOV) visualization over the simulated intersection area. Cameras are positioned at fixed ground-level locations around the cross-shaped intersection, with each frustum representing the approximate coverage area used for drone detection. This configuration simulates an urban surveillance setup for aerial threat monitoring.

### B.3.5 Environmental and Weather Control

To simulate varying visibility, occlusion, and sensor noise conditions, we configured the environment's weather and lighting settings through a structured JSON file. These parameters affect environmental factors such as fog density, cloud coverage, and material wetness, enabling realistic degradation in visual input. This is crucial for evaluating the robustness of aerial perception models under non-ideal conditions, such as limited visibility or lens obstruction.

The key parameters controlling the scene conditions include:

- **Time of day**: Determines sun position and lighting angle, affecting shadows, brightness, and ambient illumination.

- **Cloud coverage**: Controls the density of cloud layers, influencing overall light diffusion and scene contrast.

- **Rain**: Simulates precipitation and visual distortion due to water droplets on the camera lens.

- **Fog**: Introduces atmospheric occlusion, reducing scene visibility and simulating low-contrast environments.

These parameters are parsed and applied at runtime, allowing fine-grained control over environmental conditions in each data capture session. Figure 12 shows a representative sample of scenes under different configurations, illustrating the variability introduced by our weather simulation.

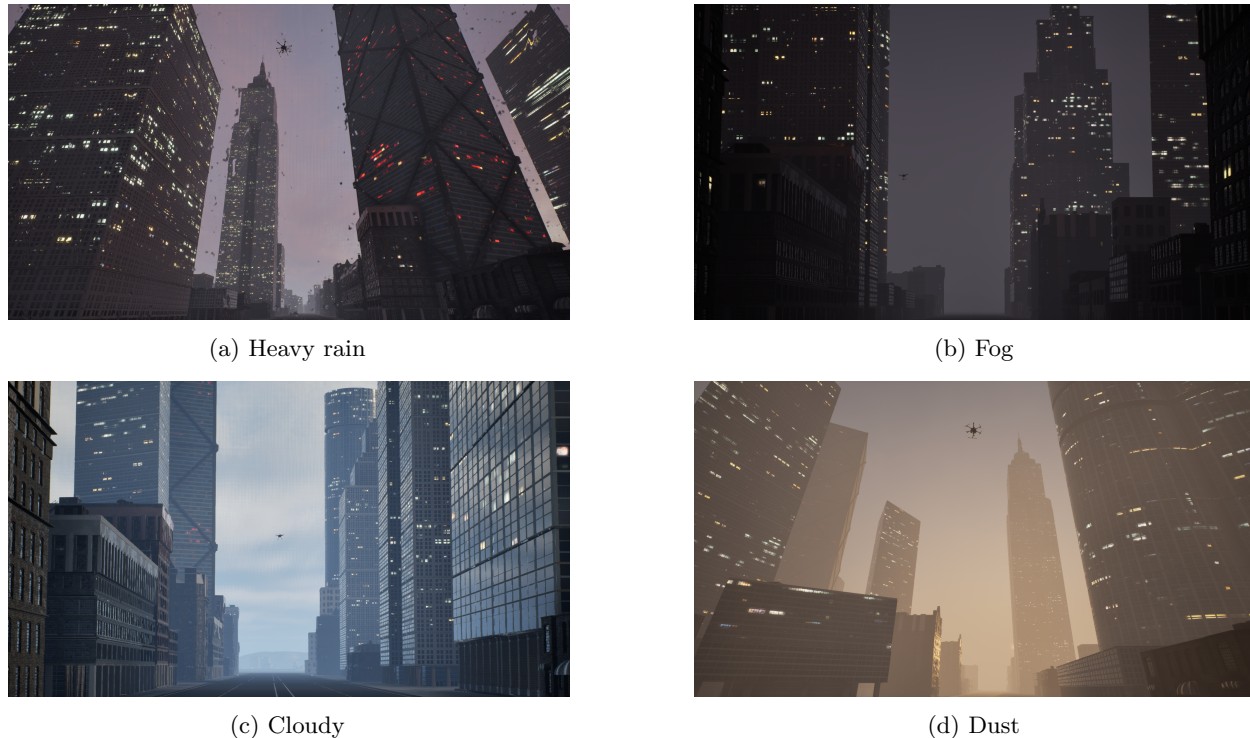

(a) Heavy rain                (b) Fog

(c) Cloudy                (d) Dust

Figure 12: Example views of the simulated environment under different weather conditions captured from different camera sensors. Each condition introduces unique visibility and sensor challenges for aerial perception.

The complete dataset contains camera specifications with location encoded in $x$, environment conditions in $z$ and the number of way-points covered (in red) as measured by drone detections for each $x, z$ pair, which is the utility value $f(x, z)$. Additionally, we also have access to utility values for set-context pairs $f(X, z)$, which will be used to train the surrogate network.

### B.3.6 Training

We train the embedding network $\phi(x, z)$ using singleton utilities $f(x, z)$ obtained from the oracle using contrastive loss with log-determinant regularization, and the surrogate $\rho$ using mean squared error on $\mathcal{D}_{\text{real}}$ and submodularity + monotonicity regularization on $\mathcal{D}_{\text{synth}}$. Both networks are optimized using Adam with learning rate scheduling and early stopping.

Table 4: Hyperparameters and Network Architecture for Contrastive Learning of Embeddings ($\phi$)

| Component | Details |
|---|---|
| **Embedding Model Architecture** | |
| *Input Dimension* | 12 (8 sensor features + 4 environmental factors) |
| *Intermediate Layers* | Dense(64, activation=ReLU) |
| *Output Dimension* | $d = 8$ (softplus activation) |
| *Embedding Split* | $k = 6$ (utility), $d - k = 2$ (independence) |
| | |
| **Training Hyperparameters** | |
| *Learning Rate* | $10^{-3}$ |
| *Batch Size* | 32 |
| *Number of Epochs* | 30 |
| *Triplet Margin* | $\delta_{\text{margin}} = 0.1$ |
| *Independence Loss Weight* | $\lambda_{\text{indep}} = 1$ |
| | |
| **Optimizer** | Adam Optimizer |

Table 5: Hyperparameters and Network Architecture for Surrogate Set Function ($\rho$)

| Component | Details |
|---|---|
| **Set Function Model Architecture** | |
| *Input Dimension* | Embedding Size $= 8$ |
| *Hidden Layers* | Dense(256, activation=ReLU), Dense(128, activation=ReLU) |
| *Output Layer* | Dense(1, activation=sigmoid) |
| | |
| **Training Hyperparameters** | |
| *Learning Rate* | Initial $10^{-4}$, Exponential Decay |
| *Decay Steps* | 10 |
| *Decay Rate* | 0.2 |
| *Number of Epochs* | 30 |
| *Loss Weights* | $\lambda_{\text{mono}} = 0.1$, $\lambda_{\text{sub}} = 0.1$ |
| | |
| **Optimizer** | Adam Optimizer with Learning Rate Schedule |

### B.4 Experimental Details for Personalized Document Summarization

In this section, we describe the experimental setup for evaluating our framework on a personalized document summarization task based on the Reuters Corpus Lewis et al. (2004). This task is formulated as a contextual subset selection problem, where the goal is to select a set of articles that best summarizes user-specific topical preferences.

### B.4.1 Data Extraction and Preprocessing

The Reuters Corpus consists of newswire articles annotated with multiple topics. We preprocess the dataset using Latent Dirichlet Allocation (LDA) with $n_{\text{topic}} = 10$ topics. Each article $x \in \mathcal{X}$ is then represented by a topic distribution vector

$$P(i|x), \qquad i \in \{1, \ldots, n_{\text{topic}}\},$$

where $P(i|x)$ denotes the probability that article $x$ is associated with topic $i$.

**User Feature Extraction ($z$):**  A user is represented by a topic preference vector $z \in \Delta^{n_{\text{topic}}}$, i.e., a probability simplex over topics. This vector models the user's affinity towards different topics. We sample 10 points on this simplex uniformly at random to form the context set (i.e., set of users) $\mathcal{Z}$.

**Article Feature Extraction ($x$):**  Each article is represented by its 10-dimensional topic distribution $P(\cdot|x) \in [0,1]^{10}$ obtained from LDA. We randomly sample 200 articles from the corpus to form the ground set $\mathcal{V}$.

**Utility Label Generation ($f(S,z)$):**  For a user $z$ and subset of articles $S \subseteq \mathcal{V}$, the personalized coverage utility is defined as

$$f(S, z) = \sum_{i=1}^{n_{\text{topic}}} z_i \left( 1 - \prod_{x \in S} (1 - P(i|x)) \right).$$

This is a monotone submodular function measuring how well $S$ covers the user's topical preferences.

**Triplet Generation:**  For training the embedding network, we construct triplets $(x_a, x_p, x_n)$ relative to a user $z$:

- $x_a$: an article aligned with one of the user's top-3 preferred topics,

- $x_p$: an article aligned with one of the user's top-5 preferred topics,

- $x_n$: an article aligned with one of the user's least 2 preferred topics.

**Set Utility Dataset:**  For each user $z$, we generate 1000 subsets $S$ with size 3–7, and compute utilities $f(S, z)$. These serve as training and evaluation data for the surrogate set function model.

### B.4.2 Network Architectures and Hyperparameters

**Training Details:**  The embedding network is first trained with the triplet and independence losses. The learned embeddings are then frozen and provided to the surrogate set function model, which is trained to minimize MSE loss with additional regularizers enforcing submodularity and monotonicity. Experiments are repeated with varying train/test splits (70/30, 80/20), and we report generalization performance compared against DeepSets and LeaSuRe. Our framework consistently yields lower test loss across different training sizes.

Table 6: Hyperparameters and Network Architecture for Contrastive Learning of Embeddings ($\phi$)

| Component | Details |
|---|---|
| **Embedding Model Architecture** | |
| *Input Dimension* | 20 (10 topics + 10 user preference weights) |
| *Intermediate Layers* | Dense(64, activation=ReLU) |
| *Output Dimension* | $d = 8$ (softplus activation) |
| *Embedding Split* | $k = 6$ (utility), $d - k = 2$ (independence) |
| | |
| **Training Hyperparameters** | |
| *Learning Rate* | $10^{-3}$ |
| *Batch Size* | 32 |
| *Number of Epochs* | 50 |
| *Triplet Margin* | $\delta_{\mathrm{margin}} = 0.1$ |
| *Independence Loss Weight* | $\lambda_{\mathrm{indep}} = 1$ |
| | |
| **Optimizer** | Adam Optimizer |

Table 7: Hyperparameters and Network Architecture for Surrogate Set Function ($\rho$)

| Component | Details |
|---|---|
| **Set Function Model Architecture** | |
| *Input Dimension* | Embedding Size = 8 |
| *Hidden Layers* | Dense(128, activation=ReLU), Dense(64, activation=ReLU) |
| *Output Layer* | Dense(1, activation=sigmoid) |
| | |
| **Training Hyperparameters** | |
| *Learning Rate* | Initial $10^{-5}$, Exponential Decay |
| *Decay Steps* | 10 |
| *Decay Rate* | 0.2 |
| *Number of Epochs* | 50 |
| *Loss Weights* | $\lambda_{\mathrm{mono}} = 0.5$, $\lambda_{\mathrm{sub}} = 0.5$ |
| | |
| **Optimizer** | Adam Optimizer with Learning Rate Schedule |

