# OpenReview forum: "Learning Structured Set Utility Functions with Contrastive Element Representations"
_TMLR — Decision pending for TMLR_

### Review · Reviewer_QQ1E · 2026-02-26

**Summary Of Contributions:**

This paper introduces SELECT (SEt function Learning framework via Contrastive Element representations), a framework designed to learn contextual set utility functions that exhibit monotone submodular structures.

The core innovation lies in a two-part architecture:

- Contrastive Element Embedding Network: Maps high-dimensional elements and contexts into a low-dimensional latent space.
- Aggregation & Surrogate Network: Employs a Deep Sets-style architecture to predict set-level utility from aggregated embeddings.

The authors provide a theoretical convergence analysis for their regularised contrastive loss under projected gradient descent. They evaluate SELECT on diverse tasks, such as movie recommendation, sensor selection, and document summarisation, demonstrating superior utility prediction and downstream greedy subset selection performance compared to standard baselines like Deep Sets.

**Audience:**

Yes

**Audience Explanation:**

Yes. This work will interest researchers in submodular optimisation, representation learning, and set function learning. The integration of contrastive learning with structural set-function properties is a timely contribution, especially for practitioners dealing with "black-box" or context-dependent utility functions in recommendation systems and sensor networks.

**Broader Impact Concerns:**

There are no broader impact concerns.

**Claims And Evidence:**

Yes

**Claims Explanation:**

Yes. The claims are supported by a combination of rigorous theoretical proofs and empirical evaluations.

**Requested Changes:**

- In Section 4.1 (Remark 1), you mention that injective representations can be achieved when d < n by assigning a dedicated coordinate with unique positive values. It would be beneficial to clarify if this specific coordinate-assignment trick was used in the experiments, where d=8 and n might be larger, e.g., n=50 for MovieLens.
- While Equation 5 is presented as a volume-maximizing regularizer, its connection to the principle of maximum likelihood could be more explicitly discussed in the appendix to strengthen the statistical justification of the loss.
- While Algorithm 1 details the contrastive part, the paper would benefit from a more cohesive description of how the two stages interact in practice. Specifically, clarifying whether the embedding network phi is frozen" or if there is any end-to-end fine-tuning during the aggregator rho training would be helpful, as Section B.1.2 suggests freezing but Section 10 mentions an "end-to-end framework”.
- In Figure 2e, SELECT is compared to an "Oracle Greedy.” It would be better to include a random selection baseline (i.e., a baseline that performs a random guess to select subsets).
- There are two papers [1,2] very related. It would be beneficial to discuss the connection to them.

[1] Ou, Zijing, et al. "Learning neural set functions under the optimal subset oracle." *Advances in Neural Information Processing Systems* 35 (2022): 35021-35034.

[2] Xie, Binghui, et al. "HORSE: hierarchical representation for large-scale neural subset selection." *Advances in Neural Information Processing Systems* 37 (2024): 4852-4877.

---

> ### Author Response · Authors · 2026-05-13
>
> >In Section 4.1 (Remark 1), you mention that injective representations can be achieved when $d < n$ by assigning a dedicated coordinate with unique positive values. It would be beneficial to clarify if this specific coordinate-assignment trick was used in the experiments, where $d=8$ and $n$ might be larger, e.g., $n=50$ for MovieLens.
>
> We thank the reviewer for highlighting this subtle but important technical point. We appreciate the careful reading of the manuscript.
>
> Indeed, in the experimental settings where $d<n$, we employed the coordinate-assignment construction described in Remark 1 to ensure injectivity of the representations. We have now explicitly clarified this missing implementation detail in Appendix B.

---

> ### Author Response · Authors · 2026-05-13
>
> > While Algorithm 1 details the contrastive part, the paper would benefit from a more cohesive description of how the two stages interact in practice. Specifically, clarifying whether the embedding network phi is frozen" or if there is any end-to-end fine-tuning during the aggregator rho training would be helpful, as Section B.1.2 suggests freezing but Section 10 mentions an "end-to-end framework”.
>
> We thank the reviewer for pointing out the need for a clearer description of how the two stages interact in practice.
>
> There are, in principle, two natural ways to couple the embedding network $\phi$ and the aggregator $\rho$:
>
> 1. **Two-stage training with frozen embeddings.** First, $\phi$ is trained using the contrastive objective (including the log-determinant regularizer). After convergence, the embeddings are frozen and used as fixed representations for training $\rho$ in the second stage.
>
> 2. **End-to-end fine-tuning.** One may pre-train $\phi$ using the contrastive objective, initialize $\rho$, and then jointly fine-tune both $\phi$ and $\rho$ under the downstream objective in an end-to-end manner.
>
> In our framework, we adopt the first approach. Specifically, we train $\phi$ using the contrastive objective, freeze the resulting embeddings, and then train $\rho$ on top of these fixed representations. This separation is deliberate in our theoretical analysis as well -- the theoretical guarantees we establish for representation diversity and injectivity apply directly to the embeddings produced in the first stage and are preserved when $\phi$ is frozen.
>
> We agree that this was not stated with sufficient clarity in the main text, and that we used the term "**end-to-end**", while our framework was not doing the end-to-end fine-tuning. We have revised the wordings in the manuscript to explicitly clarify that we propose a **2-stage** framework for contextual set function learning. The end-to-end fine-tuning variant is a natural and interesting extension, but it is not explored in the current work and remains an avenue for future work.

---

> ### Author Response · Authors · 2026-05-13
>
> > While Equation 5 is presented as a volume-maximizing regularizer, its connection to the principle of maximum likelihood could be more explicitly discussed in the appendix to strengthen the statistical justification of the loss.
>
> We thank the reviewer for their insightful observation and the suggestion to clarify the statistical interpretation of Equation (5).
>
> Equation (5) contains the regularizer $\log \det(UU^\top),$ where $U$ denotes the matrix of embedding vectors produced by the network $\phi$. This term promotes geometric diversity by maximizing the volume of the parallelepiped spanned by the embeddings.
>
> **1. Maximum-Likelihood Interpretation**
> If the embeddings are viewed as samples from a zero-mean Gaussian distribution, $e_i = \phi(x_i) \sim \mathcal{N}(0,\Sigma),$ then the log-likelihood of $m$ samples is
>
> $\ell(\Sigma)=-\frac{m}{2} \log \det(\Sigma)-
> \frac{1}{2} \sum_{i=1}^m e_i^\top \Sigma^{-1} e_i+ \text{constant}.$
>
> Under this model, the empirical covariance estimator satisfies
>
> $\hat{\Sigma} = \frac{1}{m} U^\top U.$
>
> Evaluating the likelihood at the maximum-likelihood estimate yields a term proportional to $-\log \det(U^\top U),$ up to additive constants. Thus, optimizing $\log \det(UU^\top)$ can be interpreted as maximizing the likelihood under a Gaussian covariance model.
>
> Equivalently, since the differential entropy of a Gaussian satisfies $H(\mathcal{N}(0,\Sigma)) =\frac{1}{2} \log \det(\Sigma)+\text{constant},$ the regularizer promotes a maximum-entropy (maximum-volume) configuration of embeddings.
>
> **2. Distribution-Free Geometric Interpretation**
>
> If no underlying distributional assumption is imposed, the regularizer admits a purely geometric interpretation.
>
>  We note that $\det(U^\top U)= \prod_{i=1}^{d} \sigma_i^2,$ where $\sigma_i$ are the singular values of $U$. Hence, $\log \det(U^\top U)=2 \sum_{i=1}^{d} \log \sigma_i.$
>
> Maximizing this quantity prevents spectral collapse by ensuring that no singular direction degenerates. Equivalently, it maximizes the generalized variance of the embeddings and increases the volume of the representation space. This promotes independence and diversity among embeddings without requiring any probabilistic modeling assumption.
>
> We have included a subsection (A.7) in the appendix to explicitly articulate both interpretations, strengthening both the statistical and geometric justification of the regularizer.

---

> ### Author Response · Authors · 2026-05-13
>
> > In Figure 2e, SELECT is compared to an "Oracle Greedy.” It would be better to include a random selection baseline (i.e., a baseline that performs a random guess to select subsets).
>
> We thank the reviewer for this constructive suggestion. We agree that including a random selection baseline would  help quantify the absolute gain over uninformed selection for interpreting the performance of SELECT.
>
> We have updated Figure 2e to include a random selection baseline and report its average performance over multiple trials.

---

> ### Author Response · Authors · 2026-05-13
>
> > There are two papers [1,2] very related. It would be beneficial to discuss the connection to them.
> >
> > [1] Ou, Zijing, et al. "Learning neural set functions under the optimal subset oracle." Advances in Neural Information Processing Systems 35 (2022): 35021-35034.
> >
> > [2] Xie, Binghui, et al. "HORSE: hierarchical representation for large-scale neural subset selection." Advances in Neural Information Processing Systems 37 (2024): 4852-4877.
>
> We thank the reviewer for pointing us to the related works [1] and [2]. Both papers study neural network–based approximations of set-valued functions and present elegant, well-designed modeling frameworks for subset selection. However, our problem formulation differs in several meaningful ways, particularly in terms of the learning assumptions, the availability and type of supervision samples, the incorporation of structural properties, and the alignment with downstream optimization objectives. Below, we clarify these distinctions to more precisely position our contributions relative to [1] and [2].
>
> In [1], the authors formulate the learning problem using a maximum likelihood estimation (MLE) framework:
>
> \\[
> \\begin{array}{ll}
> \\arg\\max\_{\\theta} & \\mathbb{E}\_{\\mathbb{P}(V,S)}[\\log p\_{\\theta}(S \\mid V)] \\\\
> \\text{s.t.} & p\_{\\theta}(S \\mid V) \\propto F\_{\\theta}(S;V), \\; \\forall S \\in 2^V .
> \\end{array}
> \\]
>
> This approach models the full conditional distribution $p_\theta(S \mid V)$ over subsets. Once such a model is trained, one can recover an optimal subset for a given ground set via likelihood maximization, without explicitly enforcing or requiring structural properties such as submodularity.
>
> In [2], the authors propose HORSE, a hierarchical attention mechanism to model rich interactions among elements for large-scale neural subset selection. While attention is expressive, it inherently introduces quadratic interaction complexity and additional computational overhead, which can become significant as the ground set grows. The proposed hierarchical partitioning strategy alleviates this to some extent, but it alters the interaction structure by decomposing the set into groups, potentially limiting full cross-group dependencies.
>
> Implicit in the formulations of both [1] and [2] is access to supervised samples $\\{(V\_i, S\_i^\*)\\}\_{i=1}^N$, where $S\_i^\* \\subseteq V\_i$ represents an optimal (high quality) subset for the ground set $V_i$. However, this does not explicitly account for the cardinality bounds of these subsets $S_i$, i.e., if one is only interested to identify $S_i \subseteq V_i, s.t. |S_i| \le k$. Both $V_i$ and $k$ can change arbitrarily and this makes the task of learning the optimal subset more complex. In many practical settings, obtaining such labeled (optimal) subset data can be challenging. While certain domains may naturally provide these samples, generating them in general requires repeatedly solving combinatorial optimization problems, many of which are NP-hard.
>
> Furthermore, because the likelihood involves normalization over all subsets $S \in 2^V$, scalability becomes a concern as the size of the ground set increases. In contrast, our approach avoids explicit normalization over the full power set and is designed to scale efficiently while remaining directly aligned with the downstream subset selection objective.
>
> Additionally, we make a key departure from these works by considering contextual set functions, where the utility depends not only on the subset but also on the associated context. Neither [1] nor [2] address contextual set function learning. Our objective is to learn set function approximators that generalize across unseen subsets and unseen contexts, while simultaneously supporting efficient subset selection under such varying conditions.
>
> We added a discussion of these related works in Section 1 of our revised manuscript.

---

### Review · Reviewer_Su7E · 2026-02-28

**Summary Of Contributions:**

The paper introduces a representation‑based learning approach for evaluating monotone submodular utility functions, with a clear focus on generalizing effectively to elements or contexts that were not seen during training. It also seeks to advance the broader objective of developing a more efficient method for selecting high‑quality subsets.

In what follows, the strengths and weaknesses associated with the paper are stated:

The following summarizes the paper’s main strengths and weaknesses.

Strengths:
- The work addresses a broad class of set utility functions, particularly context‑dependent ones, making it applicable to modern deep learning models such as CLIP, vision–language models, and diffusion models.
- The introduction of spectral regularization to mitigate dimensional collapse is a notable and conceptually appealing contribution.
- Theoretical justification behind the embedding-related loss training is provided.

Weaknesses:
- In Algorithm 1, sampling x_a uniformly may fail to draw elements from small or sparsely populated clusters, potentially producing a poorly diversified selected set.
- Several notational choices are insufficiently defined, which disrupts the reading flow; for example, the term $J^z(\theta _{\phi })$ in Assumption 2 is not clearly introduced.
- The experimental section lacks ablation studies exploring the influence of key hyperparameters $k$ and $\alpha$.
- The method does not currently allow incremental addition of new elements or contextual variables without full retraining or fine‑tuning.

**Audience:**

Yes

**Audience Explanation:**

The paper focuses on learning utility functions that capture how individual elements matter on their own and how they contribute within a set of elements. This formulation underlies tasks such as subset selection, feature selection, and a range of related applications where evaluating the value of a set is essential, which would also help in the advancement of submodular-based coresets, where utility functions can be learnt and used over sets of elements, rather than using data-agnostic utility functions.

**Broader Impact Concerns:**

There are no such concerns.

**Claims And Evidence:**

Yes

**Claims Explanation:**

Partially. As mentioned in the weaknesses above, there is still room for improvement, as well as clarifications that need to be made.

**Requested Changes:**

In what follows, a list of questions is provided in addition to the weaknesses that you need to address:
* Having anchors being sampled uniformly at random seems like a striking weakness, especially when the data contains densely large clusters and sparsely much smaller ones as well. Did you consider taking anchors differently? For example, using coresets (submodular, sensitivity-based, or even heuristically-based coresets), or $D^2$-sampling? One coreset that can be used here with practical modifications would be [1], for example.
* By removing $\mathcal{L}_{\text{mono}}$, would we model a non-monotone submodularity utility function?
* Can you add experiments measuring the approximability of your surrogate model in having a submodular function? Is that possible?
* How did you decide on $k$ and $\alpha$? can you add ablation concerning these parameters to better grasp their effect on the robustness and effectiveness of your proposed framework?
* In Algorithm 2, how did you decide $N_s$?
* In Figure 2(a): this figures show that that only the first 2 are unique while the rest are almost meaningless. Is this a result of the dataset itself? If so, can you run on a different dataset where the singular values do not plummet to almost non-existent by the 10th singular value or 20th singular value?


_______________________________
[1] Jubran, I., Tukan, M., Maalouf, A., & Feldman, D. (2020, November). Sets clustering. In International Conference on Machine Learning (pp. 4994-5005). PMLR.

---

> ### Author Response · Authors · 2026-05-13
>
> > In Algorithm 1, sampling x\_a uniformly may fail to draw elements from small or sparsely populated clusters, potentially producing a poorly diversified selected set.
>
> > Having anchors being sampled uniformly at random seems ... One coreset that can be used here with practical modifications would be [1], for example.
> >
> > [1] Jubran, I., Tukan, M., Maalouf, A., \& Feldman, D. (2020, November). Sets clustering. In International Conference on Machine Learning (pp. 4994-5005). PMLR.
>
> We thank the reviewer for this insightful suggestion regarding coreset-based anchor sampling. We agree that uniformly sampling the anchor $x_a$ may underrepresent small or sparsely populated regions of the dataset when the data distribution contains clusters of highly unequal sizes. We also thank the reviewer for suggesting references to consider for diversity-aware sampling mechanisms.
>
> In response, we have incorporated a **coreset-based anchor pool** constructed using **farthest-first traversal** [1]. This is a simple and effective way to promote diversity while remaining computationally lightweight and easy to implement in our framework. For the ground set $\mathcal{V}=\\{x_1,\ldots,x_n\\}$ with element features $x_i \in \mathbb{R}^d$, we iteratively construct a coreset $C \subseteq \mathcal{V}$. The procedure begins by selecting one element at random, and then repeatedly adds the element that is farthest from the current coreset in feature space:
> $$
> x_{t}=\arg\max_{x_i\in\mathcal{V}} \min_{x_j\in C_{t-1}} \|x_i-x_j\|_2 .
> $$
> One heuristic to determine the size of the coreset is $|C| = \sqrt{\mathcal{V}}$, which can provide sufficient diversity coverage across the ground set.  Anchors are then sampled from this coreset $C$ instead of uniformly from $\mathcal{V}$. We compare the spread of anchor pool in the feature space and the prediction quality (test loss) with respect to uniform anchor sampling baseline and observe a better spread of anchor elements and slight improvments in test loss. The new experiments are presented in Section B.2 - Ablation Studies of the revised manuscript.
>
> [1] Jubran, I., Tukan, M., Maalouf, A., \& Feldman, D. (2020, November). Sets clustering. In International Conference on Machine Learning (pp. 4994-5005). PMLR.

---

> ### Author Response · Authors · 2026-05-13
>
> > Several notational choices are insufficiently defined, which disrupts the reading flow; for example, the term $J^z(\theta_{\phi})$ in Assumption 2 is not clearly introduced.
>
> We thank the reviewer for pointing this out. We agree that some notational choices, such as $J^{z}(\theta_{\phi})$ in Assumption 2, were not introduced with sufficient clarity. We have revised the manuscript to ensure that all notation is clearly defined when first introduced, improving the overall readability and flow of the presentation.

---

> ### Author Response · Authors · 2026-05-13
>
> > By removing $\\mathcal{L}\_{mono}$, would we model a non-monotone submodular utility function?
>
> We thank the reviewer for this question. The term $\\mathcal{L}\_{\\text{mono}}$ enforces monotonicity by penalizing violations of the condition $f(A) \\leq f(B)$ for $A \\subseteq B$. If $\\mathcal{L}\_{\\text{mono}}$ is removed, this constraint is no longer enforced and the learned function need not be monotone. The remaining regularization $\\mathcal{L}\_{\\text{sub}}$ can still softly encourage submodular structure; however, it does not guarantee monotonicity.
>
> More broadly, these regularizers are introduced to recover the closest neural network approximation of the underlying true utility within a chosen hypothesis class. When both regularizers are present, the model searches within a hypothesis class of **monotone submodular functions**. If $\\mathcal{L}\_{\\text{mono}}$ is removed, the search space expands to the broader class of **submodular functions**, which may include both monotone and non-monotone functions. Consequently, monotonicity may no longer be present.

---

> ### Author Response · Authors · 2026-05-13
>
> >Can you add experiments measuring the approximability of your surrogate model in having a submodular function? Is that possible?
>
> We thank the reviewer for this helpful suggestion.
>
> - One practical way to assess this is through downstream subset selection performance, which we already evaluate in our experiments in Section 6.2. Specifically, we run greedy maximization on the learned surrogate and then evaluate the resulting subsets using the ground-truth oracle utility. If the surrogate is both accurate and approximately submodular, the greedy solutions induced by the surrogate should transfer well to the true utility. Thus, this experiment provides a decision-relevant notion of approximability of the learned surrogate.
>
> - Second, one can directly evaluate violations of the diminishing-returns property. This can be done by sampling tuples $(A,B,e)$ with $A \subseteq B$ and $e \in \mathcal{V} \setminus B$, and checking whether
> $f(A \cup \{e\}) - f(A) \geq f(B \cup \{e\}) - f(B)$. We repeat this over many sampled tuples and varying the number of test samples, and report the percentage of violations (the average magnitude of violations). This will quantify how closely the learned surrogate satisfies submodularity. We have now added this evaluation and present this study in Section B.2 - Ablation Studies of the revised manuscript.

---

> ### Author Response · Authors · 2026-05-13
>
> > The experimental section lacks ablation studies exploring the influence of key hyperparameters   $k$  and $\alpha$.
>
> > How did you decide on $k$  and $\alpha$ ? can you add ablation concerning these parameters to better grasp their effect on the robustness and effectiveness of your proposed framework?
>
> We thank the reviewer for this suggestion. The hyperparameters $k$ and $\alpha$ were selected based on preliminary validation experiments to ensure stable training and good empirical performance. We agree that a more systematic study would further clarify their influence on the robustness and effectiveness of the framework. Accordingly, we have included an ablation study varying $k$ and $\alpha$ and report the results in Section B.2 - Ablation Studies of the revised manuscript.

---

> ### Author Response · Authors · 2026-05-13
>
> > In Algorithm 2, how did you decide $N_s$  ?
>
> We thank the reviewer for the question. The parameter $N\_s$ denotes the number of synthetic samples used to expose the monotonicity and submodularity violations during training. In our implementation, we set $N\_s = 2|\mathcal{D}\_{real}|$, where $\mathcal{D}\_{real}$ is the real supervision dataset. This provides a sufficient number of synthetic samples for the regularization terms while keeping the contribution balanced relative to the real data. In this sense, $N\_s$ effectively controls the strength of the regularization signal during training. We have clarified this choice in Appendix B of the revised manuscript.

---

> ### Author Response · Authors · 2026-05-13
>
> > In Figure 2(a): this figures show that only the first 2 are unique while the rest are almost meaningless. Is this a result of the dataset itself? If so, can you run on a different dataset where the singular values do not plummet to almost non-existent by the 10th singular value or 20th singular value?
>
> We acknowledge the reviewer for this observation. As the reviewer noted, this behavior is largely driven by the structure of the dataset itself, where only a small number of directions capture most of the variation in the learned representation. This leads to a rapid decay in the singular value (eigenvalue) spectrum. In addition, this effect is also caused by the embedding dimensionality  $d$ and the size of the ground set of elements $|\mathcal{V}|$.  As a result, in the matrix $U^\top U$, only few  singular values can meaningfully capture variation, while the remaining ones are necessarily close to zero. After a few dominant directions, the spectrum quickly becomes negligible.
>
> We run the experiment on a different dataset (by sampling a different set of movies and users) and report the resulting eigenvalue spectrum which captures slightly more meaningful variation before the spectrum fully decays. The plots are presented in Section B.2 - Ablation Studies of the revised manuscript.

---

### Review · Reviewer_y1Dq · 2026-04-30

**Summary Of Contributions:**

*Summary:* The authors propose a framework for learning context-dependent monotone sub-modular utility functions over sets. A stylized version of their feature learning phase is shown to converge to a stationary point. Finally, the authors perform empirical experiments to assess their framework's ability to generalize across contexts and sets.

*Main Strengths:*
- The idea of learning contextual sub-modular set functions is interesting and worth pursuing.

*Main Weaknesses:*
- The authors compare their framework only to ablated versions thereof. There is no comparison to other solution methods in the literature.
- The ablated versions are systematically disadvantaged, as they effectively train on less training data. The proposed method trains on labelled instances during feature learning and during utility function learning, whereas the ablated versions only train in one of those phases. It is not clear if the performance gains hold beyond this advantage.
- There is no discussion about how to choose the hyper-parameters $d,k$ and $\epsilon$, determining the feature dimension, the number of "diversity dimensions" and the similarity threshold respectively. As these are novel aspects of the proposed framework, it is not clear what effects these parameters have and make the framework itself opaque.
- The contribution of the theoretical results is unclear. The analyzed algorithm does not match their proposed algorithm in many aspects. Smoothness assumptions, smoothed version of the loss, decreasing similarity threshold and projected gradient descent are all key elements for the main theorem to hold. Non of these aspects is actually implemented in the proposed feature learning module. Beyond that, the proposed learning framework as a whole is not investigated.
- There is no discussion about how many true labels are required for the embedding network or the utility network to work efficiently well. A critical aspect to asses whether this more sophisticated architecture is efficient enough. This does not align with the claim that their framework "offers a principled and scalable approach for learning surrogate set functions ... with *limited* supervision sample".

**Audience:**

No

**Audience Explanation:**

The results do not allow to make conclusions about the efficiency of the framework. In particular, it is not clear how the framework is effectively applied, as critical hyper-parameters are not discussed.

**Broader Impact Concerns:**

No concerns.

**Claims And Evidence:**

No

**Claims Explanation:**

The paper fails to empirically compare their framework to other work in the literature. The ablated versions are systematically disadvantaged, leaving no ground for conclusions about sample efficiency. As sample efficiency is not clear, it is not clear to what extend the framework can handle expensive to evaluate set functions. Novel aspects of their framework, requiring the selection of hyper-parameters, is not discussed.

**Requested Changes:**

*Changes that are critical for acceptance:* All main weaknesses.

*Additional Weaknesses that would strengthen:*
- The paper claims to address situations in which utility functions are expensive to evaluate, unknown or lack an explicit analytical form. Two out of three of their experiments consider the case in which the true utility function is efficiently computeable and known for a large subset of context-item pairs. This true utility function is used to label thousands of subsets.
- The experiments lack the important ablation to the following method: Estimate the part of the utility function that requires generalization to unseen test points. This approach might be much more sample efficient as it basically boils down to classical supervised learning.
- The need for feature learning is motivated by the situation in which the set elements are high dimensional and hence difficult to handle. Yet, in the experiments, the input dimension is at most 24.
- There is no discussion about how the feature learning module handles large sets. The diversity regularization becomes quickly intractable in this case.

---

> ### Author Response · Authors · 2026-05-13
>
> > The authors compare their framework only ... in the literature.
>
> >
> > The ablated versions are systematically disadvantaged ... gains hold beyond this advantage.
>
> We thank the reviewer for raising this important concern. We would like to clarify that the baselines we used are not merely ablated versions of our framework, but are established methods from the literature that are directly applicable to the problem setting we consider.
>
> Specifically, **Deep Sets** is a standard permutation-invariant architecture for learning set functions, and **LeaSuRe** is a prior framework for learning surrogate utility functions with monotonicity and submodularity regularization. Both methods can be easily modified to incorporate contextual information by taking the context vector $z$ as part of the model input along with $x$. Therefore, these baselines represent relevant existing solution methods that can be adapted/used for our problem setting.
>
> The reason we focus on these baselines is that many other methods in the literature do not directly fit our setting. In particular, our task requires learning a surrogate utility function $\hat f(S,z)$ from oracle-labeled set-context pairs, generalizing to unseen subsets and unseen contexts, while preserving permutation invariance. Several existing methods either require optimal subset labels, optimize a non-contextual objective, or are designed for a different supervision model. Hence, **Deep Sets** and **LeaSuRe** are the most direct literature baselines for the learning-based contextual set-function approximation task studied in this paper.
>
> Regarding the reviewer’s concern about the amount of supervision, we agree that SELECT uses singleton utility feedback during representation learning in addition to set-level labels during utility learning. To isolate whether the gains arise from the proposed representation-learning mechanism rather than from additional oracle feedback, we conducted an additional budget-matched ablation study.
>
> Let
> $$
> B_{\text{total}} = B_{\text{single}} + B_{\text{set}},
> $$
> where $B_{\text{single}}$ is the number of singleton utility queries used for contrastive representation learning and $B_{\text{set}}$ is the number of set-level utility queries used for learning the aggregator. In the budget-matched comparison, SELECT uses this split budget, while the baselines are given the same total number of oracle-labeled samples $B_{\text{total}}$ directly for set-level utility learning. This ensures that all methods receive the same total amount of labelled oracle feedback. The new experiments are presented in Section B.2 - Ablation Studies of the revised manuscript.
>
> The results show that SELECT continues to outperform the baselines under this matched-budget setting, indicating that the improvement is not simply due to using more training data. Rather, the singleton labels used in the contrastive stage provide a structured utility-aware representation that improves generalization across unseen sets and contexts. We include this clarification along with the budget-matched ablation study in Appendix B.2 of the revised manuscript.

---

> ### Author Response · Authors · 2026-05-13
>
> > There is no discussion about how to choose the hyper-parameters $d,k$ and $\epsilon$, ... make the framework itself opaque.
>
> We thank the reviewer for pointing this out. We agree that the manuscript should better explain the role and selection of the hyperparameters $d$, $k$, and $\epsilon$.
>
> In our framework, $d$ controls the total embedding dimension and therefore the expressive capacity of the element representation. The parameter $k$ controls how many dimensions are used for the contrastive utility-similarity component, while the remaining $d-k$ dimensions provide additional diversity capacity through the independence regularization. The threshold $\epsilon$ controls the construction of positive and negative pairs in the triplet dataset: smaller $\epsilon$ enforces stricter utility similarity, while larger $\epsilon$ produces more positive pairs but may weaken the semantic precision of the contrastive signal.
>
> In our experiments, these parameters were selected through validation to ensure stable training and good prediction performance. We agree with the reviewer that a more systematic explanation would improve transparency. As also noted in response to another reviewer, we have included an ablation study varying $k$ and $\alpha$; we have extended this discussion to include $d$ and $\epsilon$ as well, and clarified their practical effect on representation quality and utility prediction. The new experiments and discussions are presented in Section B.2 - Ablation Studies of the revised manuscript.

---

> ### Author Response · Authors · 2026-05-13
>
> > The contribution of the theoretical results is unclear. The analyzed algorithm ... as a whole is not investigated.
>
> We acknowledge the reviewer for raising this concern. We would like to clarify that the theoretical analysis is intentionally developed under certain assumptions to make the nonconvex contrastive learning objective mathematically tractable. The smoothness assumptions, compact parameter space, and projected gradient descent are standard analytical tools used to establish convergence of the regularized contrastive representation-learning stage. They are not meant to exactly replicate every implementation detail, but to provide a principled justification that the proposed log-determinant regularized objective/loss admits stable convergence behavior under controlled conditions.
>
> We also clarify that the decaying parameter in the theorem is not the triplet similarity threshold $\epsilon$ used for constructing positive and negative pairs. It is the numerical regularization parameter $\varepsilon_t$ inside the log-determinant term.
>
> In practice, our implementation does not explicitly enforce all of these assumptions. Nevertheless, the experiments show stable training and effective learning of utility-aware representations without requiring the smoothed surrogate. In this sense, the empirical results provide a stronger practical message: even though the theory is proved under certain assumptions, the proposed feature-learning module remains stable and effective in the actual implementation, where such assumptions are not fully satisfied.
>
> While our theory focuses on the representation learning aspect, leveraging the two-phase training, the guarantees of the full framework are characterized by the learned surrogate's structural properties. Once the surrogate is learned to be approximately submodular (to be more precise, additive weak-submodular) through the regularization terms, subset selection using greedy algorithm directly inherits the standard near-optimality guarantees from the submodular optimization literature. Thus, our theoretical results support the representation-learning component, while the downstream subset selection guarantees follow from existing results for (weak) submodular optimization litertaure [1].
>
> [1] Alieva, Ayya, Aiden Aceves, Jialin Song, Stephen Mayo, Yisong Yue, and Yuxin Chen. "Learning to make decisions via submodular regularization." In International Conference on Learning Representations. 2020.

---

> ### Author Response · Authors · 2026-05-13
>
> > There is no discussion about how many true labels ... with limited supervision sample".
>
> We thank the reviewer for raising this important point. We would like to clarify that our experiments in Sections 6.1, 6.2 and 6.3 already evaluate this aspect through varying train-test splits.
>
> Specifically, we vary the percentage of labelled subset samples used for training and evaluate the test loss on held-out, unseen subsets. Under each split, SELECT and all baselines are given the same percentage of labelled subset-utility samples for training. Therefore, the comparison directly measures how efficiently each method uses the same amount of supervision.
>
> The results show that, for the same number of subset-level labels, SELECT achieves lower test loss than the baselines. This indicates that SELECT predicts utilities of unseen sets more accurately under limited supervision. Equivalently, to reach the same test loss achieved by SELECT, the baselines would require more labelled subset samples.
>
> Thus, the claim of learning with limited supervision is supported by the train-test split experiments: to reach the same test loss, and therefore the same prediction quality on unseen subsets, SELECT requires fewer training samples than Deep Sets or LeaSuRe.

---

> ### Author Response · Authors · 2026-05-13
>
> > The paper claims to address situations in which utility functions are expensive to evaluate, unknown ... label thousands of subsets.
>
> We thank the reviewer for raising this point. We agree that the recommendation and document summarization experiments use known analytical utility functions. The proposed framework assumes access to samples from a utility oracle, where the oracle may be an analytically computable function, a simulator, or a data-driven routine that provides the corresponding utility values. These settings are used because they allow us to generate controlled labels efficiently, perform extensive train-test split experiments, and directly evaluate whether the learned surrogate accurately predicts utilities on unseen subsets and contexts. In particular, using inexpensive-to-evaluate oracles enables large-scale simulation studies that validate the proposed framework before applying it to more expensive or domain-specific utility evaluation settings.
>
> Additionally, these tasks (recommendation and document summarization) are also standard benchmark experiments used in the closest related literature on submodular utility learning and subset selection. Including them helps demonstrate that our framework is applicable beyond a single domain and can be evaluated across a wide variety of contextual set-function learning tasks.
>
> At the same time, the sensor selection experiment directly reflects the motivating setting of our framework. In that task, we do not have a closed-form analytical utility function for a set of sensors. Instead, the utility is obtained by running a high-fidelity simulator and measuring the resulting sensor coverage under different environmental conditions. Re-running this simulation for every candidate sensor set and context is precisely the expensive oracle-evaluation setting that our framework aims to address.
>
> Thus, these experiments serve complementary purposes. Recommendation and summarization provide controlled benchmark evaluations, while sensor selection demonstrates the practical expensive-oracle setting where a learned surrogate is especially useful.

---

> ### Author Response · Authors · 2026-05-13
>
> > The experiments lack the important ablation to the following method: Estimate the part of the utility function that requires generalization to unseen test points. This approach might be much more sample efficient as it basically boils down to classical supervised learning.
>
> We thank the reviewer for this suggestion. We agree that, when the utility function admits a known decomposition, one could learn only the unknown component and then plug it into the analytical utility expression. However, this approach is not directly applicable to the general setting considered in our work. Our framework is designed for cases where the set utility may be expensive, unknown, or may not have an explicit analytical form. In such cases, there may not be a “known part” of the utility function that can be isolated and learned through classical supervised learning.
>
> For instance, in the sensor selection setting, the set utility is obtained from simulator-based coverage measurements and does not come with a closed-form decomposition into known or unknown parts that can be be reduced to standard supervised learning over individual components. Thus, while the suggested approach is valuable for special cases where the utility structure admits such structure/decomposition, it does not directly address the broader contextual set-function learning problem considered in this paper.

---

> ### Author Response · Authors · 2026-05-13
>
> > The need for feature learning is motivated by the situation in which the set elements are high dimensional and hence difficult to handle. Yet, in the experiments, the input dimension is at most 24.
>
> We thank the reviewer for pointing this out. We would like to clarify that the motivation for feature learning in our framework is not solely dimensionality reduction.
>
> High-dimensional inputs are one motivating case, but the broader goal is to learn **context-conditioned, utility-aware element representations**. In particular, the embedding network $\phi(x,z)$ is designed to represent elements based on how they contribute to utility under a given context, rather than only compressing raw features.
>
> Thus, even when the raw feature dimension is moderate, feature learning remains useful because it organizes elements in a latent space according to contextual utility similarity. The low-dimensional representation is an added benefit, but the main purpose is to obtain representations that improve generalization to unseen sets and contexts.
>
> We will clarify this motivation in the revised manuscript to avoid giving the impression that feature learning is needed only when the raw element features are high dimensional.

---

> ### Author Response · Authors · 2026-05-13
>
> > There is no discussion about how the feature learning module handles large sets. The diversity regularization becomes quickly intractable in this case.
>
> We thank the reviewer for raising this concern. We would like to clarify that the diversity regularization is not inherently intractable, since its computation can be controlled through the embedding dimension and stabilized through the $\epsilon_t I$ term in the log-determinant objective/loss.
>
> In practice, the regularizer is applied to the embedding matrix, not to the full power set of subsets. Moreover, when the number of elements is large and the embedding dimension satisfies $d \ll n$, the log-determinant term can be computed in the lower-dimensional embedding space using the equivalent Gram formulation, which reduces the dominant computation from depending cubically on the number of elements to depending cubically on the embedding dimension [1].
>
> Additionally, for very large ground sets, there are existing works [2-3] that address large-scale representation learning efficiently, e.g., the same regularizer can be estimated over mini-batches or sampled anchor pools during training, which is standard for scalable representation learning. The role of the regularizer is to stabilize the learned representation and prevent collapse, not to require exhaustive computation over all subsets or all possible element combinations.
>
> We have clarified this scalability aspect in the Section 4 of the revised manuscript.
>
> [1] Drineas, Petros, Michael W. Mahoney, and Nello Cristianini. "On the Nyström Method for Approximating a Gram Matrix for Improved Kernel-Based Learning." Journal of Machine Learning Research, 6, no. 12 (2005).
>
> [2] Chen, Ting, Simon Kornblith, Mohammad Norouzi, and Geoffrey Hinton. "A simple framework for contrastive learning of visual representations." International Conference on Machine Learning, pp. 1597-1607. PMLR, 2020.
>
> [3] Zbontar, Jure, Li Jing, Ishan Misra, Yann LeCun, and Stéphane Deny. "Barlow twins: Self-supervised learning via redundancy reduction." In International Conference on Machine Learning, pp. 12310-12320. PMLR, 2021.

---

### Author Response · Authors · 2026-05-13
**Revised Manuscript and Rebuttal**

We sincerely thank the reviewers for their insightful and constructive comments. We have carefully incorporated all comments and made revisions to the paper to improve its clarity, positioning, technical justification, and experimental discussion. Specifically, we have included new experimental studies and ablations as requested by the reviewers along with some additional clarifications and discussions to make the theoretical results more clear to the readers. The updated portions of the manuscript are highlighted in colored text for ease of review, and we provide detailed responses to each reviewer comment below.

---

### Decision · Action_Editor_4xCP · 2026-06-29

**Recommendation:** Accept with minor revision

**Additional Comments:**

The submitted paper was reviewed by three experts which unanimously recommended the acceptance of the paper (2x leaning accept, 1x accept). In particular, the reviewers agree that the decision criteria of TMLR are met and that the presented work is in principal broadly applicable. However, the reviews also highlighted that there was a need to improve the paper regarding the experimental evaluation (ablations, baselines) and clarity, e.g., regarding some theoretical aspects. The authors have already updated the paper in that regard.

Nevertheless, there is one last aspect that should be fixed before acceptance: Please use citations properly throughout the paper. Either have them as part of sentences or add them in parentheses. E.g.,

  *Recent works have studied neural set function approximation, including EquiVSet by Ou et al. (2022), which learns neural set functions using an optimal subset oracle, and HORSE by Xie et al. (2024), which uses hierarchical attention-based representations for large-scale subset selection.*

or

   *Recent works have studied neural set function approximation, including EquiVSet (Ou et al., 2022), which learns neural set functions using an optimal subset oracle, and HORSE (Xie et al., 2024), which uses hierarchical attention-based representations for large-scale subset selection.*

**Audience:**

Yes

**Audience Explanation:**

Set utility functions with contexts can be broadly applied, e.g., in computer vision and signal processing.

**Claims And Evidence:**

Yes

**Claims Explanation:**

The advantage of the proposed modeling are approach are demonstrated in various experiments and several ablations and sensitivity analyses are conducted. There is also a theoretical analysis of the proposed approach (in a stylized setting).